# Direct observation of chaperone-modulated talin mechanics with single-molecule resolution

Soham Chakraborty[1,2], Deep Chaudhuri[1,2], Souradeep Banerjee[1,2], Madhu Bhatt[1] & Shubhasis Haldar [1✉]

Talin as a critical focal adhesion mechanosensor exhibits force-dependent folding dynamics and concurrent interactions. Being a cytoplasmic protein, talin also might interact with several cytosolic chaperones; however, the roles of chaperones in talin mechanics remain elusive. To address this question, we investigated the force response of a mechanically stable talin domain with a set of well-known unfoldase (DnaJ, DnaK) and foldase (DnaKJE, DsbA) chaperones, using single-molecule magnetic tweezers. Our findings demonstrate that chaperones could affect adhesion proteins' stability by changing their folding mechanics; while unfoldases reduce their unfolding force from ~11 pN to ~6 pN, foldase shifts it upto ~15 pN. Since talin is mechanically synced within 2 pN force ranges, these changes are significant in cellular conditions. Furthermore, we determined that chaperones directly reshape the energy landscape of talin: unfoldases decrease the unfolding barrier height from 26.8 to 21.7 $k_BT$, while foldases increase it to 33.5 $k_BT$. We reconciled our observations with eukaryotic Hsp70 and Hsp40 and observed their similar function of decreasing the talin unfolding barrier. Quantitative mapping of this chaperone-induced talin folding landscape directly illustrates that chaperones perturb the adhesion protein stability under physiological force, thereby, influencing their force-dependent interactions and adhesion dynamics.

[1] Department of Biological Sciences, Ashoka University, Sonepat, Haryana, India. [2] These authors contributed equally: Soham Chakraborty, Deep Chaudhuri, Souradeep Banerjee. ✉email: shubhasis.haldar@ashoka.edu.in

Cells interpret and respond to complex mechanical environments through focal adhesion (FA)-mediated responses that help to transduce mechanical cues into biochemical signals, supporting diverse biological processes such as cell development, migration, and proliferation[1]. These mechanical communications, through the large adhesion complex, are tightly regulated by the physical linkages and elasticity of mechanosensing proteins including intermediate filaments and multidomain proteins[2–4]. Talin is a key mechanosensitive scaffold protein that connects transmembrane integrin to the actin cytoskeleton, and mechanically tunes diverse interactions by force transmission[2–5]. Talin rod segment (R1–13) exhibits conformational changes and hierarchical unfolding of each rod domain under increasing mechanical tension[6–8]. Among them, R3 is the least stable domain that not only unfolds at ~5 pN but also exhibits equilibrium folding–unfolding transitions on a sub-second timescale by a single actomyosin contraction[2,8,9]. This force-dependent folding dynamics allow them to tune their interactome profile: such as, Rap1-interacting adapter molecule (RIAM) binds to folded R3 domain and protects from mechanical unfolding, whereas force above 5 pN results in stepwise unfolding of the domain with subsequent vinculin binding. This mutual exclusive interaction allows talin to acts as a mechanochemical switch, shifting from an initial integrin activator to the active mediator of adhesion maturation and stabilization[2,10]. Interestingly molecular chaperones, involved in the various folding-related process starting from protein translation to degradation, could play pivotal roles in FA stabilization by colocalizing with different cytoskeletal and adhesion proteins[11–18]. Since Hsp70/Hsp40 are ubiquitous cytoplasmic chaperones, they interact with different cytosolic substrates. Thus, talin being a critical mechanosensitive cytosolic protein, could potentially interact with these chaperones. However, there is no direct evidence on how these chaperones could mechanically influence talin folding dynamics and thereby, its force-dependent interactions during the adhesion maturation process.

To address this question, single-molecule magnetic tweezers have been used, which executes both force-ramp and force-clamp methodologies[19]. The force-ramp methodology, by increasing or decreasing the force at a constant rate, probes unfolding and refolding events of a protein; while applying the force-clamp method, a constant force can be applied to the protein, allowing to detect of their thermodynamics and kinetic properties under equilibrium condition. Owing to the advantage of a wide force regime of 0–120 pN with the sub-pN resolution, we are able to observe chaperone interactions both at folded and unfolded states, independently. In addition, the flow chamber allows us to introduce chaperones, individually or in combination, to probe their effect on the folding dynamics of a single protein in real-time. Finally, implementing this methodology, the force can be specifically applied to the client protein while the chaperones remain unperturbed.

Here, we systematically investigated the chaperone effects on the mechanical stability of talin R3 domain, with a set of model chaperones: holdase, foldase, and mechanically neutral. Our result showed that DnaK as a holdase chaperone mechanically weakens the R3-IVVI domain, reducing its unfolding force from 10.8 to 6 pN; whereas oxidoreductase enzyme DsbA as a foldase chaperone increases its stability by increasing the force to 14.3 pN. However, the mechanically neutral DnaKJE chaperone complex does not exhibit any additional effect on the unfolding and refolding force of the substrate. To generalize our hypothesis, we further studied the effect of Hsp70 and Hsp40, the eukaryotic homologs of DnaK and DnaJ system, on the talin mechanical stability. Interestingly, we observed that Hsp70 and Hsp40, very similar to DnaK and DnaJ, function as unfoldases, decreasing the

unfolding force of talin from 10.8 pN to ~7.9 pN, respectively. In addition, from the force-induced equilibrium condition, we are able to explicitly demonstrate the altered-mechanical response by illustrating the quantitative mapping of the chaperone-induced energy landscape. We observed chaperones could reshape the mechanical folding landscape of client proteins by changing the height of their free energy barrier without significantly affecting the transition state distance. For example, DnaJ or DnaK stabilizes the unfolded state by decreasing the unfolding barrier height, whereas DsbA stabilizes the folded state of protein by tilting the energy landscape towards the opposite pathway. We explored our data by Bell-like equation to resolve the quantitative description of the chaperone-modulated free energy landscape, which has not been reported before. From a broader viewpoint, this mechanism may have a generic mechanistic insight of how talin response and their binding kinetics with other interactors under force, could be affected by chaperone-altered stability. Overall, our result expanded the canonical function of molecular chaperones and illustrated their novel roles in modulating the stability of mechanically stretched protein substrates.

## Results

**Single-molecule folding dynamics of talin R3-IVVI domain.** Due to the presence of four threonine residues at its hydrophobic core, R3 domain is the mechanically weakest domain in talin. It unfolds earlier than any other rod domains and exhibits the equilibrium folding dynamics at ~4–6 pN[8,20]. Hence, we used a mechanically stable version of R3 domain by substituting its four threonine residues with amino acids containing larger hydrophobic residues—isoleucine and valine at 809, 833, 867, and 901 positions, referred to as R3-IVVI domain. Both the wild-type (WT) R3 and R3-IVVI domains have been extensively studied and observed to share a similar function, however, R3-IVVI exhibits amplified mechanical signature. For example, both of them bind vinculin at unfolded state but WT R3 domain unfolds at 5 pN, whereas R3-IVVI unfolds at 9 pN[10,21,22]. Since both of them exhibit similar functional properties, we used R3-IVVI domain as our substrate.

The folding–unfolding dynamics of R3-IVVI domain, at a single molecular resolution, have been observed using magnetic tweezers technology. This technology allowed the application of sub-pN level of force to understand the folding dynamics of the protein at equilibrium. This single-molecule force spectroscopy technology provides two ways for experimentation—force-ramp and force-clamp techniques. The talin construct is covalently attached to the glass surface, using the HaloTag covalent chemistry, and the C- terminal Avi-tag was biotinylated to bind the paramagnetic beads through biotin-streptavidin chemistry (Fig. 1a). Force is applied on the paramagnetic bead with a pair of permanent magnets, attached to a voice coil actuator, which controls the amount of force as an inverse function of the distance between the permanent magnet and the paramagnetic beads[23].

The force-clamp technology allowed us to provide a distinct force on the substrate and enabled us to observe its hopping between two conformations: folded and unfolded states. Figure 1b shows three representative trajectories of talin R3-IVVI folding dynamics, and the population of both the folded and unfolded states, at three distinct forces. The folding dynamics were deduced by analyzing several of these folding trajectories and dividing the relative population of the folded state by the total duration of the observed dynamics[21,22]. At 10 pN, the talin R3-IVVI domain displays an almost equal proportion of folded and unfolded conformation with an extension change of ~19 nm; while at 8.5 pN and 11.5 pN, the domain mostly occupies folded

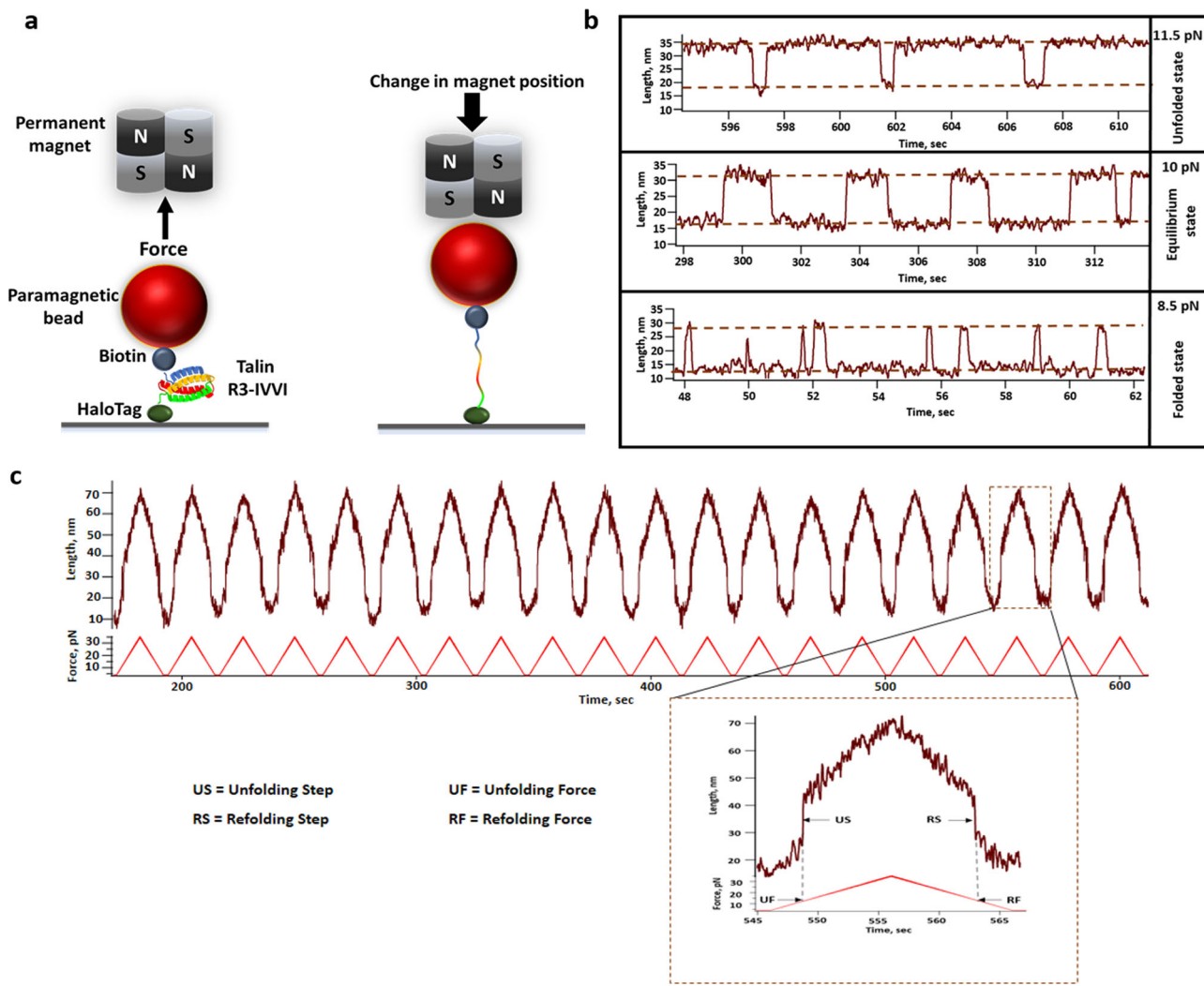

**Fig. 1 Real-time magnetic tweezers set-up to study the folding dynamics and mechanical stability of talin R3-IVVI. a** Schematic representation of magnetic tweezers experiment where the engineered construct of a single R3-IVVI domain is flanked between the glass surface and the streptavidin-coated paramagnetic bead using halotag chemistry. The applied force is controlled by changing the distance between a paramagnetic bead and a permanent magnet. **b** Force-dependent folding dynamics of talin R3-IVVI domain using magnetic tweezers where talin R3-IVVI exhibits a strong force-dependent folding dynamics, which is characterized as reversible folding–unfolding transitions, yielding extension changes of ~20 nm. At 11.5 pN force, it is mostly present in unfolded state, whereas, at 8.5 pN, it is mostly in the folded state. At 10 pN, both the states occupy almost equal populations. **c** Successive force-ramp cycle, from 4 to 35 pN at a constant loading rate of 3.1 pN/sec, are applied to observe distinct unfolding and refolding steps at particular forces. In the inset, a single force-ramp is magnified where the unfolding step (US) and refolding step (RS) have been extrapolated to the force-ramp axis to measure the unfolding force (UF) and refolding force (RF), respectively.

and unfolded population, respectively. This force-dependent folding dynamics has been observed to shift in the presence of different chaperones, which is also evident from the change in unfolding and refolding force in force-ramp experiment.

**Chaperones modulate the unfolding and refolding force of talin**. The mechanical stability of R3-IVVI was monitored using the force-ramp technology, where we performed the force-ramp experiment by increasing the force from 4 to 35 pN at a constant loading rate of 3.1 pN/s, followed by a successive force-decrease scan with the same loading rate. During each force increase scan, we observed unfolding events as sudden increases in the extension, which is also observed in the force-decrease scan as a sudden decrease in the extension at particular peak forces. We estimated the peak force by vertically joining the unfolding and refolding steps (US and RS) to the equivalent force in force-ramp axis (as shown in inset Fig. 1c). For easy understanding, we have

also shown a representative force-ramp experiment in the figure below and have described how to estimate unfolding and refolding force (UF and RF). We monitored more than 150 events to quantify the unfolding and refolding force of R3-IVVI domain (Fig. 1c). Interestingly, in the presence of chaperones, the force-extension curves are hysteric in nature due to the altered binding dynamics in chaperone-substrate interaction (Supplementary Fig. 1).

We systematically investigated the chaperone effect on the talin mechanical stability by monitoring their unfolding and refolding force using the force-ramp protocol. The unfolding force of talin has been observed to decrease with the unfoldase chaperone, if compared to the absence of the chaperones (control). For example, the unfolding force is decreased from $10.8 \pm 0.2$ pN (mean±standard error of the mean) in control, to $7.9 \pm 0.4$ pN in the presence of DnaJ and $7.8 \pm 0.5$ pN with apo-DnaK (Fig. 2a, b). We also performed the experiment with the DnaK+DnaJ complex (DnaKJ-ATP) and observed that the unfolding force is

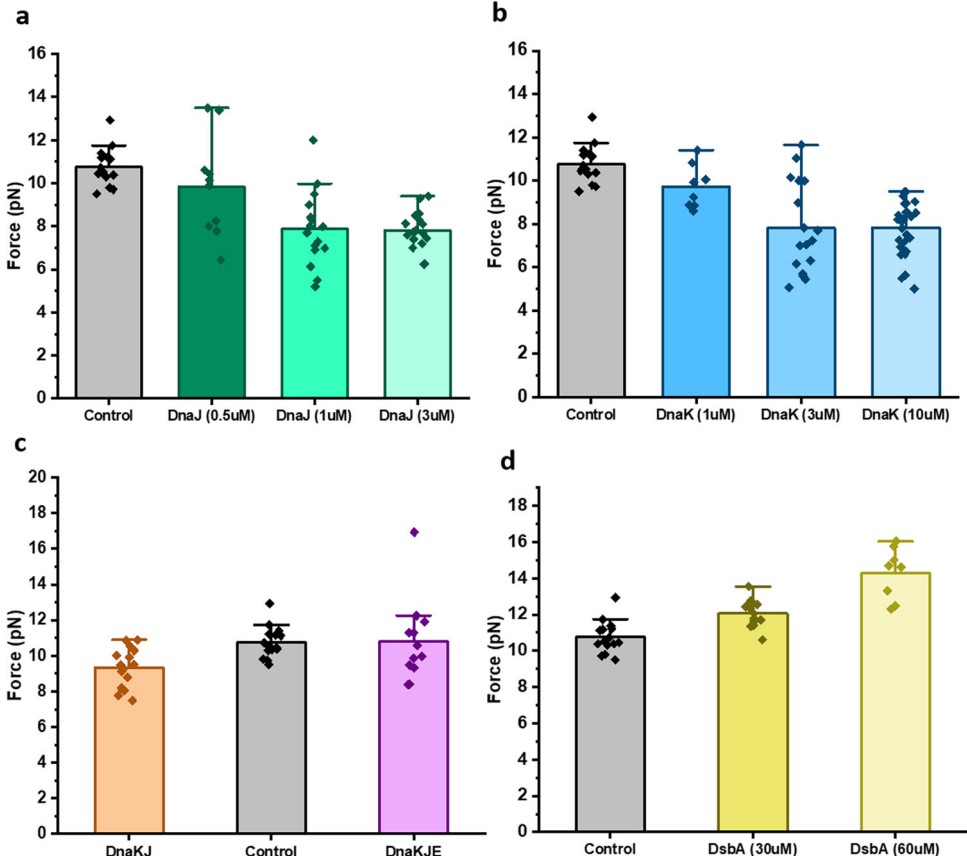

**Fig. 2 Force-ramp experiment detects the chaperone-mediated change in unfolding force of talin.** In an unfolding force-ramp protocol, the force is increased from 4 to 35 pN at a constant rate of 3.1 pN/sec. **a** Unfolding event in the absence of chaperones (control) is observed at 10.8 ± 0.2 pN, which has been observed to decrease steadily with DnaJ concentration. **b** In the presence of DnaK, talin unfolds at lower force and the force decreases with the increasing concentration of DnaK. **c** With DnaK, DnaJ, and ATP, talin unfolds at the lower force of 9.4 ± 0.3 pN, while upon addition of GrpE (DnaK+ DnaJ + GrpE+ ATP), it unfolds at a similar force with the control. These experiments have been performed with 10 mM ATP and 10 mM MgCl$_2$. The buffer is changed every 30 min with fresh ATP to keep the sufficient supply of ATP. **d** DsbA increases the unfolding force in force-ramp experiments. Talin domain unfolds at 10.8 ± 0.2 pN force, whereas it increases to 12.1 ± 0.2 pN with 30 μM DsbA and reaches 14.3 ± 0.5 pN in the presence of 60 μM DsbA. Data points are measured by averaging the unfolding forces of more than three individual molecules with $n \geq 8$ unfolding extensions. Error bars are represented as s.e.m.

decreased to 9.4 ± 0.3 pN, implying that the DnaKJ complex still works as a holdase under force (Fig. 2c). However, the unfolding force is reverted to that of control in the presence of the DnaK +DnaJ+GrpE (DnaKJE-ATP) complex (Fig. 2c). Remarkably, in the presence of 60 μM oxidized DsbA, a known force-dependent foldase chaperone[24], the mechanical stability of talin increased to 14.3 ± 0.5 pN (Fig. 2d). This chaperone-modulated mechanical stability has also been reconciled from the unfolding/rupture force analysis, where the unfolding forces are plotted against the varying loading rate from 1 to 7 pN/s and has been observed to follow Bell–Evans distribution. We observed that in the absence of any chaperones, the unfolding force at zero loading rate is 7.6 ± 0.01 pN, which decreases to 6.9 ± 0.1 pN in the presence of DnaK while increasing up to 10.4 ± 0.3 pN with DsbA, signifying the altered-mechanical stability of talin with different chaperones (Supplementary Fig. 2). Talin refolding force, similar to the unfolding force, has also been observed to alter in the presence of different chaperones. In control, the refolding force is 10 ± 0.3 pN, which decreases with the unfoldases (Fig. 3a, b) and increases in the presence of foldase like DsbA (Fig. 3d). This holdase function is also observed in the case of refolding force with DnaKJ-ATP complex, however, the addition of GrpE to this complex restores the mechanical stability in talin by increasing the force comparable to that of control (Fig. 3c).

**Statistical analysis of chaperone-modulated unfolding and refolding force of talin.** Statistical significance of the unfolding and refolding force between different chaperone data sets has been measured by one-way analysis of variance (ANOVA), followed by checking the pair- comparison at *$p \leq 0.05$ level (Supplementary Figs. 3 and 4). We observed that the unfolding force changes with concentrations for different chaperones; however, found to be not significant in all the cases. Since this mechanical effect becomes saturated at a particular chaperone concentration, the unfolding force remains unchanged and thus, no significant differences are observed beyond that concentration. For example, with DnaJ, the unfolding force becomes saturated at 1 μM concentrations and no significant differences are observed upon increasing the concentration to 3 μM (Supplementary Fig. 3a, b). Similarly, for DnaK, no such differences are observed between 3 and 10 μM concentrations (Supplementary Fig. 3c, d). Interestingly, we observed that both the unfolding and refolding force decreases significantly in the presence of the DnaKJ complex (Supplementary Figs. 3e, f, and 4e, f). Similar to unfolding force, the refolding force has also been observed to decrease and become saturated at 1 μM DnaJ and 3 μM DnaK, exhibiting the same statistical significance (Supplementary Fig. 4a–d). With 60 μM DsbA, the refolding force has been observed to significantly increase, if compared to the control (Supplementary Fig. 4g, h).

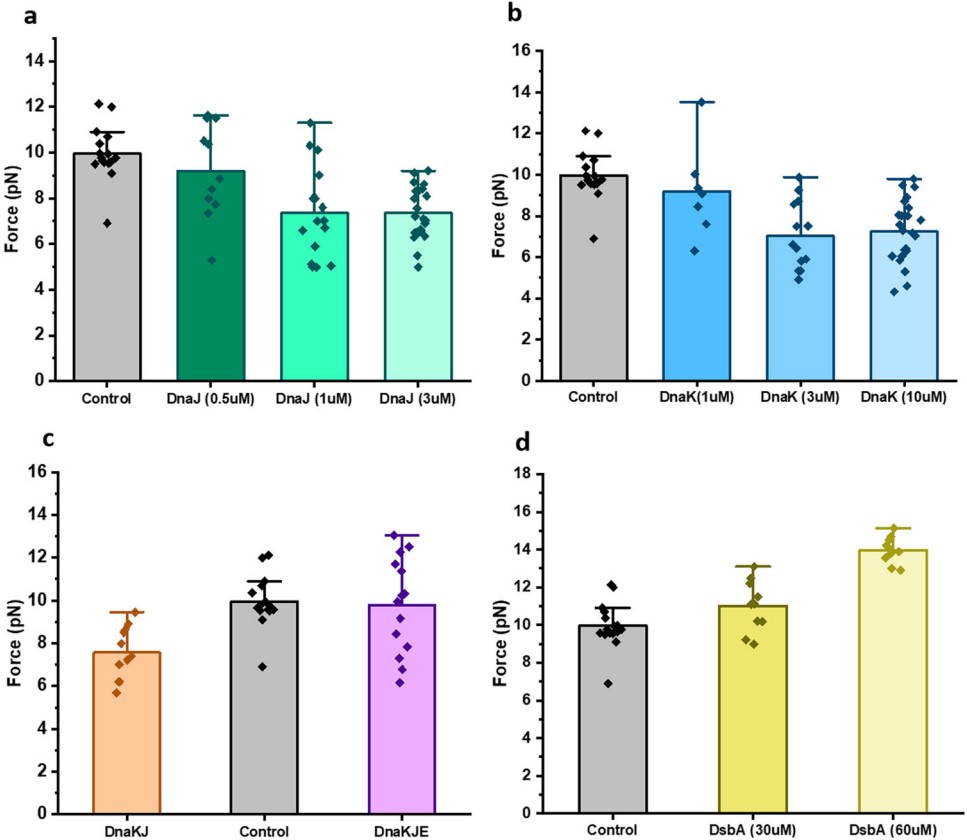

**Fig. 3 Force-ramp protocol detects the chaperone-mediated change in refolding force of talin.** In a refolding force-ramp protocol, the force is decreased from 35 pN to 4 pN at a rate of 3.1 pN/sec. **a** In the absence of DnaJ, the refolding event of talin is observed at 10 ± 0.3 pN, and with the gradual addition of DnaJ, the refolding force decreases steadily and becomes saturated at 1 μM DnaJ. **b** Similar to DnaJ, DnaK decreases the refolding force of talin domain. **c** In the presence of DnaKJ and ATP, R3-IVVI refolds at lower force (7.6 ± 0.4 pN) than control (10 ± 0.3 pN), whereas upon addition of GrpE to this mixture (DnaK, DnaJ, GrpE, and ATP), it refolds at 9.8 ± 0.6 pN. The buffer is changed every 30 min with fresh ATP to keep the sufficient supply of ATP. **d** The refolding force has been observed to increase from 10 ± 0.3 pN (control) to 11 ± 0.4 pN in the presence of 30 μM DsbA and increases further to 14 ± 0.2 pN with 60 μM DsbA. Data points are measured by averaging the refolding forces of more than three individual molecules with $n \geq 7$ refolding steps. Error bars are represented as s.e.m.

**Unfolding and refolding kinetics of talin observed under force in the presence and absence of chaperones.** Dwell time analysis at different force ranges, in the presence of different chaperones, suggest that talin optimally manifests the folding dynamics at a force, where both the folded and unfolded states are equally populated (Supplementary Figs. 5–9). This reflects its unfolding and refolding rates during chaperone interactions. We calculated the unfolding and refolding rates of talin, as described by Tapia-Rojo et al.[21]. The rates were derived from the inverse of averaged folding and unfolding dwell times ($s^{-1}$) at particular forces, as determined from the exponential fit to the dwell time distribution. From these measured rate values in mechanical chevron plots, we obtained a linear relationship between their ln values and the applied force, and fitted with a Bell-like equation (Eq. 1 and Eq. 2). Since talin exhibits folding–unfolding transitions within a small force regime of 2 pN, the data could be explained by the Bell model[21].

$$\ln k_U = \left\{ \ln A - \left( \frac{\triangle G_U^{\dagger}}{k_B T} \right) \right\} + \left( \frac{F \triangle x_U}{k_B T} \right) \quad (1)$$

$$\ln k_F = \left\{ \ln A - \left( \frac{\triangle G_F^{\dagger}}{k_B T} \right) \right\} + \left( \frac{F \triangle x_F}{k_B T} \right) \quad (2)$$

$F$ is the applied force, for which the free energy barrier is reduced by $F\triangle x$, $\triangle x$ represents the distance to the transition state

for folding, ($\triangle x_F$) and unfolding ($\triangle x_U$). $\triangle G_U^{\dagger}$ and $\triangle G_F^{\dagger}$ are the height of free energy barrier at zero force for unfolding and folding, respectively. $k_F$ and $k_U$ are the folding and unfolding rate constants, $k_B$ Boltzmann constant, $T$ is the temperature at Kelvin scale and $A$ is attempt frequency pre-factor, $10^6$ sec$^{-1}$ [25–27]. The cross-point of unfolding and refolding rates is referred to as intersection force, where talin exhibits equal unfolding and refolding rates. However, chaperone interactions perturb the intersection force of talin to different forces such as, in control, the intersection force is 9.7 pN (Fig. 4a), whereas holdase like DnaJ or apo-DnaK downshift this force to 7.9 pN and 6 pN, respectively (Fig. 4b, c). By contrast, the mechanical foldase like DsbA increases it to 14.9 pN (Fig. 4d). Owing to the pronounced mechanical effect of chaperones on talin stability, their force ranges are also observed to change significantly, which is evident from the fraction folded analysis (Supplementary Fig. 10). We also monitored the intersection force in the presence of chaperone complexes at their different nucleotide states (Supplementary Figs. 11–13).

Additionally, from the mechanical chevron plots, we determined the height of the energy barriers and transition state distance for the unfolded state, allowing us to construct a quantitative chaperone-induced free energy landscape of protein (Table 1). The protein exhibits a force-dependent unfolding transition due to the rigid folded state, and thus, the unfolding kinetics fits better to Bell-like equation[28]. However, recent

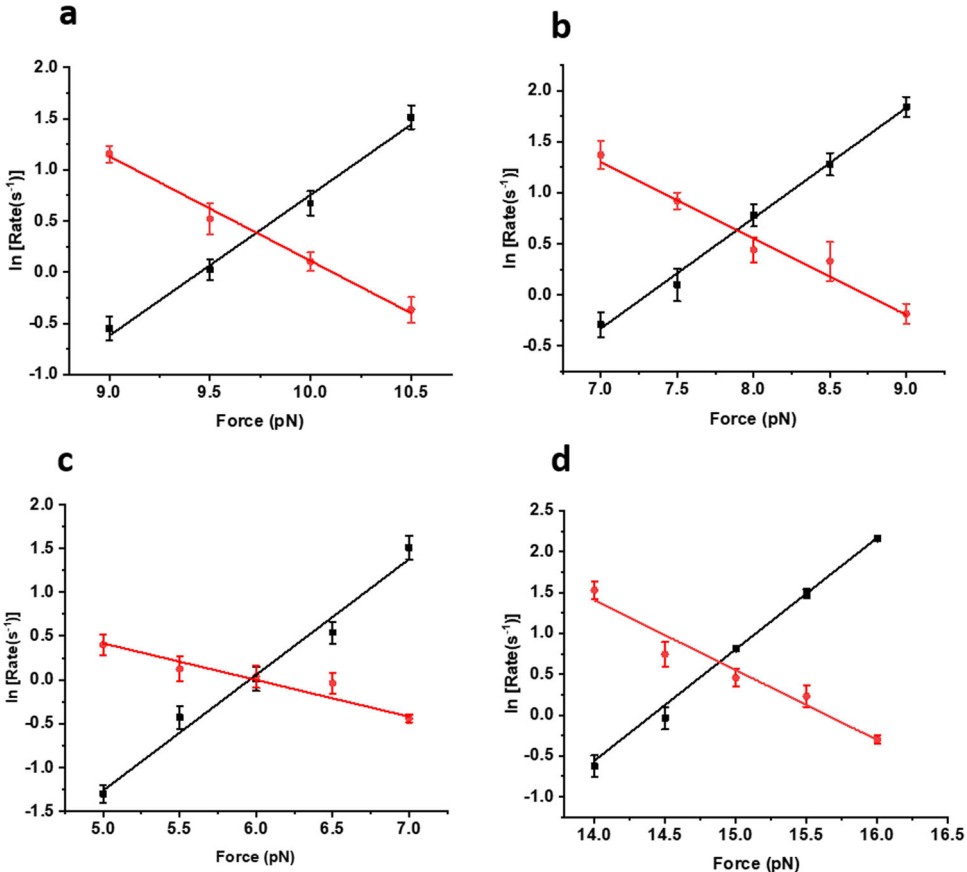

**Fig. 4 Unfolding and refolding rates of talin domain. a** Unfolding (black) and refolding (red) rates in the absence of any chaperones (control) are plotted as a function of force. **b** Effect of DnaJ on unfolding and refolding rates. Unfolding (black) and refolding (red) rates of talin in the presence of 1 µM DnaJ. Data points are calculated using more than six protein molecules. Error bars are the relative errors of log. **c** Refolding (red) and unfolding (black) rates of R3-IVVI at different forces, in the presence of 3 µM DnaK. Data points are calculated using more than five protein molecules. Error bars are the relative error of the log. **d** Unfolding (black) and refolding (red) rates of R3-IVVI are plotted against the force, in the presence of 60 µM DsbA. Data points are calculated using a minimum of three molecules per force. Error bars are the relative errors of log.

**Table 1 Unfolding kinetics parameter modulated by chaperones and their different nucleotide states.**

| | Unfolding kinetics parameters | |
| --- | --- | --- |
| | Energy barrier ($\Delta G^{\ddagger}_{U}$, $k_B T$) | Transition state distance ($\Delta x_U$, nm) |
| Control | 26.8 ± 0.8 | 1.4 ± 0.1 |
| Apo-DnaK | 21.7 ± 0.6 | 1.3 ± 0.1 |
| DnaJ | 21.7 ± 0.3 | 1.2 ± 0.1 |
| DnaK-ADP | 23 | 1.3 ± 0.1 |
| DnaK-ATP | 23 ± 2.5 | 1.2 ± 0.3 |
| DnaKJ-ADP | 22.8 ± 0.7 | 1.4 ± 0.1 |
| DnaKJ-ATP | 23.2 ± 0.1 | 1.4 |
| DnaKE-ADP | 22.8 | 1.3 |
| DnaKE-ATP | 24.4 ± 0.1 | 1.4 |
| DsbA | 33.5 ± 0.5 | 1.4 |
| Hsp70 | 23.1 | 1.3 |
| Hsp40 | 23.1 | 1.2 ± 0.3 |

The unfolding kinetics parameters are obtained by fitting the unfolding rates to Bell model (see Figs. 4 and 6). $\Delta G^{\ddagger}_{U}$ represents the height of the unfolding free energy barrier and $\Delta x_U$ is the distance from the folded state to the unfolding transition state along the reaction coordinate. Data points are calculated using more than four individual molecules per force. Errors bars are represented as s.e.m.

experimental and theoretical studies have suggested that refolding transition is strongly force-dependent due to the compliance variation of the unfolded state and collapse-associated energetics, leading to a non-linear force-dependence on the logarithmic scale, and therefore, Bell model is not a good approximation for refolding kinetics[29]. This is because unfolded states are highly flexible polypeptide chain (or soft polymer) and thus, are easily subjected to deformation by an external force, perturbing its mechanical compliance[8,29–32].

**Effect of mechanical chaperones on the WT talin domain.** To understand the effect of chaperones on the WT R3 domain, we measured the intersection force of R3-WT in the presence of different chaperones. Due to the low mechanical stability, the intersection force of R3-WT domain in the absence of any chaperones is 6.3 ± 0.3 pN. We observed holdase and foldase chaperones can change this force. For instance, in the presence of holdase chaperone such as DnaK, the intersection force decreased to 4.4 ± 0.2 pN, while the addition of foldase chaperone DsbA increased it to 9.8 ± 0.3 pN. For the DnaKJE complex, similar to R3-WT, exhibits an intersection force of 6.3 ± 0.2 pN (Supplementary Fig. 14).

**Effect of Hsp70 and Hsp40 on talin mechanical stability.** To generalize our observations with the model DnaK chaperone system, we further tested the mechanical stability of talin with the eukaryotic homologs of DnaK and DnaJ system-Hsp70 and Hsp40[33]. Since these chaperones reside in eukaryotic cytosol, they might interact with talin and thus, could change the talin folding dynamics by modulating their mechanical stability. Interestingly, we observed Hsp70 and Hsp40, very similar to DnaK and DnaJ,

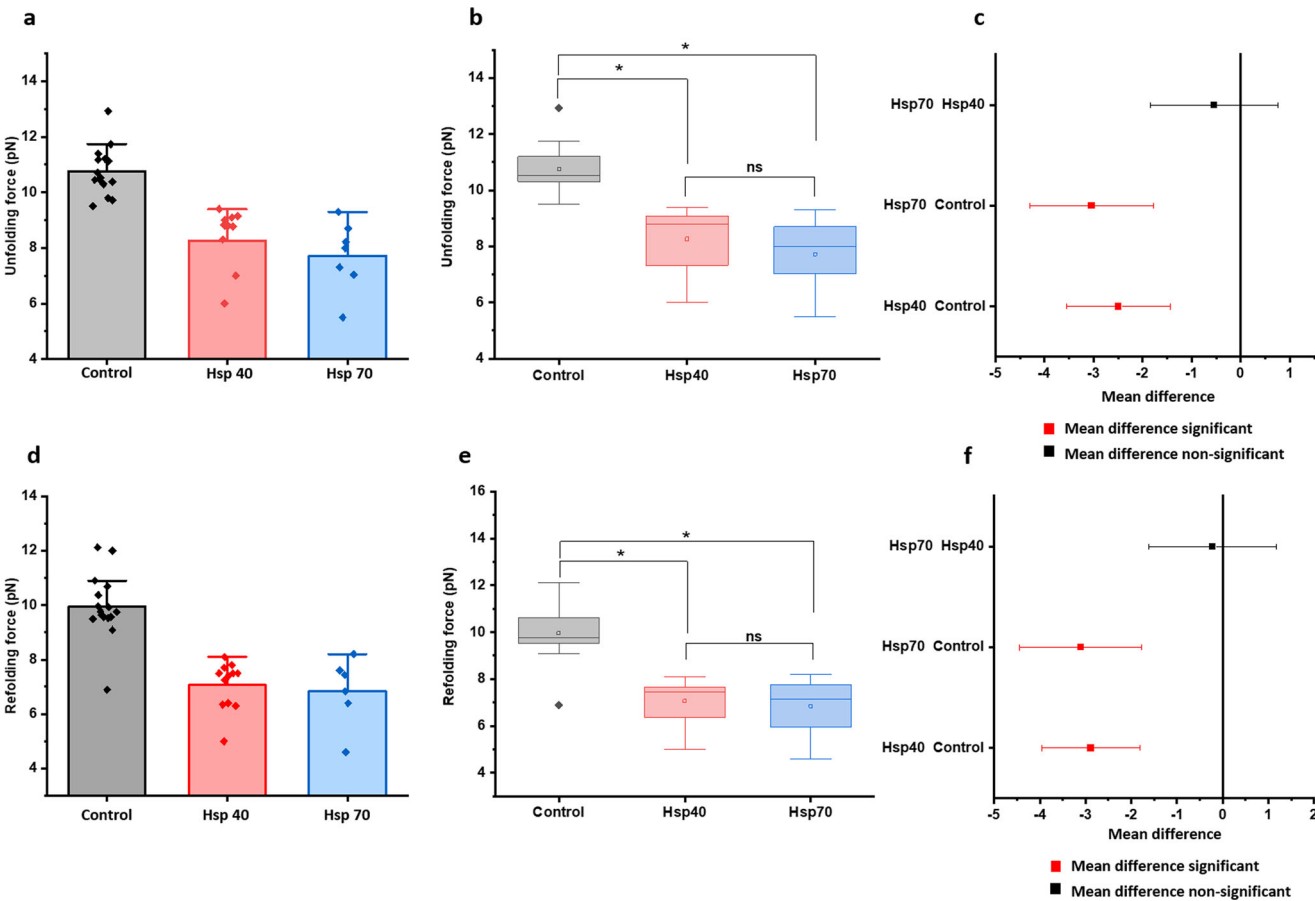

**Fig. 5 Mechanical stability of talin with Hsp40 and Hsp70.** Hsp70 and Hsp40 chaperones decrease the mechanical stability of talin by changing their unfolding and refolding force. **a** Unfolding force measured with Hsp70 and Hsp40. In the absence of any chaperones (control), the unfolding force of talin is 10.8 ± 0.2 pN, which has been observed to decrease to 8.3 ± 0.4 pN and 7.7 ± 0.5 pN with Hsp40 and hsp70, respectively. Data points are measured using more than five individual molecules with more than 12 unfolding events. Error bars represent the standard error of the mean (s.e.m.). **d** Refolding force measured with Hsp70 and Hsp40. Similarly, the refolding force has been observed to shift from 10 ± 0.3 pN (control) to 7 ± 0.5 and 6.8 ± 0.6 pN with Hsp40 and Hsp70, respectively. Data points are measured by averaging a minimum of three molecules. Error bars are represented as s.e.m. ANOVA analysis of **b**, **c** unfolding force, and **e**, **f** refolding force have been performed to check the statistical significance of the force data sets with Hsp40 and Hsp70 chaperones, followed by their pair comparison by Bonferroni post hoc test at *$p \le 0.05$ level. It has been observed that both the unfolding and refolding forces are significantly lower than that of control. Since Hsp40 and Hsp70 act as unfoldases, their mean force differences are non-significant. R-sq. = 0.630 for unfolding and 0.662 for refolding. For unfolding force analysis, $n = 15$ (Control); $n = 12$ (Hsp40); $n = 7$ (Hsp70). For refolding force, $n = 16$ (Control); $n = 12$ (Hsp40); $n = 6$(Hsp70).

act as mechanical unfoldases and reduce the mechanical stability of talin by changing their unfolding force from 10.8 ± 0.2 pN to 7.7 ± 0.5 and 8.3 ± 0.4 pN, respectively (Fig. 5a) and the unfoldase activity is also evident from the changes in refolding force (Fig. 5d). Statistical significance of both the unfolding and refolding force data has also been cross-checked by ANOVA analysis, followed by their pair comparison by Bonferroni post hoc test (Fig. 5b, c, e, f). Additionally, Hsp70 and Hsp40 have been observed to significantly shift the talin folding dynamics to the lower force regime (Supplementary Fig. 15), which has been cross-checked from the intersection force in chevron plot analysis. For example, the intersection force decreases from 9.7 pN to 7.8 pN in the presence of Hsp40 (Fig. 6a) and similarly, 7.2 pN with Hsp70 (Fig. 6b).

**Inhibition of Hsp chaperones restores talin mechanical stability.** To reconcile whether these altered talin mechanics originated due to the Hsp chaperones interaction with talin, we have performed the experiments with Hsp chaperones in the presence of their inhibitors (Figs. 7 and 8). We have used pifithrin-μ as an Hsp70 inhibitor, which is known to bind the substrate-binding

domain of Hsp70 and block its activity[34,35]. We observed that PFT-μ binding reverts the Hsp70-induced effect on the talin domain. In the presence of Hsp70, the half-point force of talin is 7.2 ± 0.4 pN, however, PFT-μ binding to Hsp70 relieves its unfoldase activity, upshifting the talin half-point force to 9.5 ± 0.7 pN (Fig. 7a). To reconcile the Hsp70-induced effect on talin mechanics, we have also performed the force-ramp experiments with Hsp70 both in the absence and presence of PFT-μ. We found that both unfolding and refolding forces increase in the presence of Hsp70 · PFT-μ complex than only Hsp70, and overlap with those of control (Fig. 7b, c). Similarly, we performed the experiments with Hsp40 in the presence of KNK437 as an Hsp40 inhibitor[36,37] and observed that Hsp40-mediated change in talin half-point force is also relieved upon the KNK437 treatment (Fig. 8a). This reversion of Hsp40 effect has also been observed in force-ramp experiment with Hsp40•KNK437 complex, where both the unfolding and refolding forces are higher, if compared to only in the presence of the Hsp40 chaperone (Fig. 8b, c).

Additionally, we performed one-way ANOVA to cross-check the statistical significance of the force data sets with Hsp chaperones and corrected their mean differences by Bonferroni

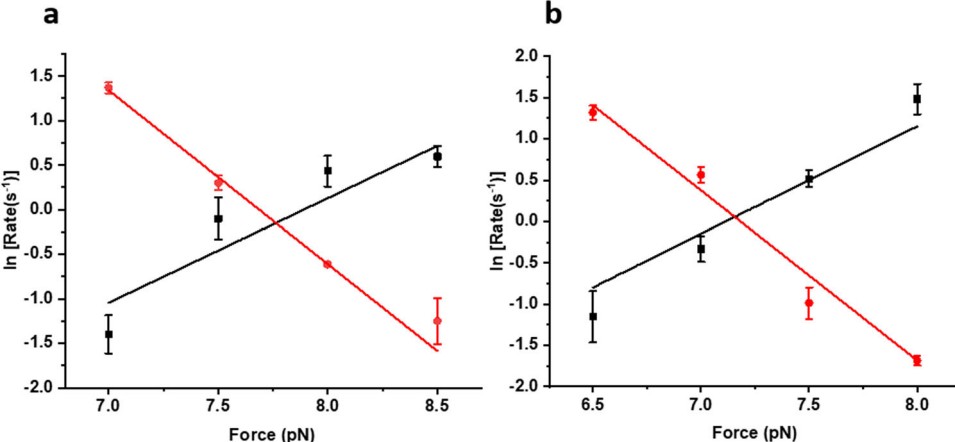

**Fig. 6 Talin kinetics with Hsp40 and Hsp70.** The unfolding and refolding kinetics are plotted as a function of force, in the presence of Hsp40 and Hsp70. Unfolding rate increases and refolding rate decreases with the force and the cross-point of these two rates is defined as intersection force. **a** The intersection force in the absence of any chaperones (control) is 9.7 pN, however, has been observed to shift to 7.8 pN with Hsp40. **b** Similarly, in the presence of Hsp70, the intersection force of talin is 7.2 pN. This certainly signifies that Hsp40 and Hsp70 decrease the mechanical stability in talin under force. Data points are calculated by an averaging minimum of three molecules per force. Error bars are the relative error of log.

post hoc test at $*p \leq 0.05$. We observed that half-point force as well as unfolding/refolding forces with Hsp70 both in the presence and absence of pifithrin-μ are statistically significant, which is also evident from a mean comparison of their data sets (Figs. 7d–i). Since PFT-μ reverts the Hsp70 activity and thereby, talin exhibits innate mechanical stability as comparable to control data; mean force differences are statistically non-significant between the Hsp70•PFT-μ complex and control. Similarly, we observed that the force range with Hsp40•KNK437 complex possesses no significant difference with that of control data, indicating that Hsp40 inhibition restores normal talin mechanics (Figs. 8d–i). These experiments strengthen our observation that change in talin mechanical stability certainly arise from the Hsp70/Hsp40 interactions and thereby, reconcile the hypothesis of the chaperone-altered-mechanical stability in talin.

## Discussion
Force response of talin domains has been well characterized using force spectroscopy and R3 domain has been observed to be the mechanically weakest domain[8], exhibiting folding–unfolding dynamics at ~5 pN. This mechanical instability is originated due to unzipping force geometry and a unique threonine belt at hydrophobic core[10]. Though R3 domain unfolds at <5 pN force, an energy cost of above 10 $k_BT$ is required for the unfolding of the domain due to the large unfolding extension of ~19 nm[38]. Interestingly, chaperones could modulate the mechanical response of the talin domains, however, the energy landscape that underlies these force-dependent interactions still remains elusive. Till now, the detailed molecular mechanism of diverse protein-chaperone interactions has been illustrated by different groups[39–45], nevertheless, no studies have elucidated the effect of chaperones in modulating the mechanical stability of force-sensing proteins in FA, which could impact the FA dynamics under physiological force regime.

The real-time magnetic tweezers technology, using force-clamp methodology, allowed us to monitor the chaperone activity on the mechanical stability of talin. We observed that apo-DnaK reduces the mechanical stability of the talin domain by decreasing the unfolding force to ~6 pN and acts as a strong unfoldase. This observation is consistent with the "closed" DnaK conformation in the apo state, which has a higher affinity for the peptide substrate, stabilizing it at unfolded state. However, this unfoldase activity is relieved upon the ATP addition, increasing the unfolding force to

7.6 pN. This indicates a lower affinity of DnaK for the peptide substrate in an 'opened' confirmation during the ATP-bound state[46]. This ATP-bound DnaK can be cycled between two states: either it remains ATP-bound or its ATP hydrolyzes into ADP; and this cycle is regulated by two other co-chaperones, DnaJ and GrpE[33,47–49]. This nucleotide cycle allows DnaK binding to the unfolded substrate in both the weaker and tighter binding manners. We observed that the addition of DnaJ to the DnaK-ATP complex again decreased the unfolding force of substrate protein, reducing its mechanical stability. It is well-known that DnaJ facilitates ATP hydrolysis[50], nevertheless, the rationale behind the decreased mechanical stability in the presence of DnaKJ-ATP complex is difficult to ascertain. Interestingly, we monitored the mechanical stability in the presence of only DnaJ and observed that the unfolding force is 7.9 pN. Thus, a decrease in the mechanical stability is probably caused by the ATP hydrolysis induced by DnaJ. To reconcile this, we also measured the mechanical stability of talin domain in the presence of DnaKJ-ADP and observed that the unfolding force is again decreased to ~6.3 pN. Therefore, from these observations, it is evident that ATP binding facilitates the substrate release from the DnaK, resulting in relatively higher unfolding force, while ATP hydrolysis is important for stabilizing the unfolded substrate within the closed DnaK, leading to lower unfolding force. We further added GrpE to both DnaK-ATP and DnaK-ADP complexes and observed a similar pattern in the unfolding force- higher unfolding force in the presence of DnaK, GrpE, and ATP and lower force in the presence of GrpE and DnaK-ADP, signifying the accelerated ATPase activity of DnaK in the presence of GrpE[51], even higher than in the presence of DnaKJ complex with either ATP or ADP. Finally, upon the addition of the DnaKJE complex with ATP, we observed the unfolding force is the same as in the absence of any chaperones, indicating that the complex does not exhibit any additional effect on the unfolding force. These forces are confirmed from the intersection force by mechanical chevron plot analysis, where the unfolding-refolding rates have been plotted against varying forces (Supplementary Figs. 11–13).

Furthermore, we observed that although chaperones are not able to change the transition state distance significantly, they reshape the free energy landscape by changing the height of the unfolding energy barrier with respect to the folded state. The holdase chaperone DnaJ stabilizes the unfolded state by

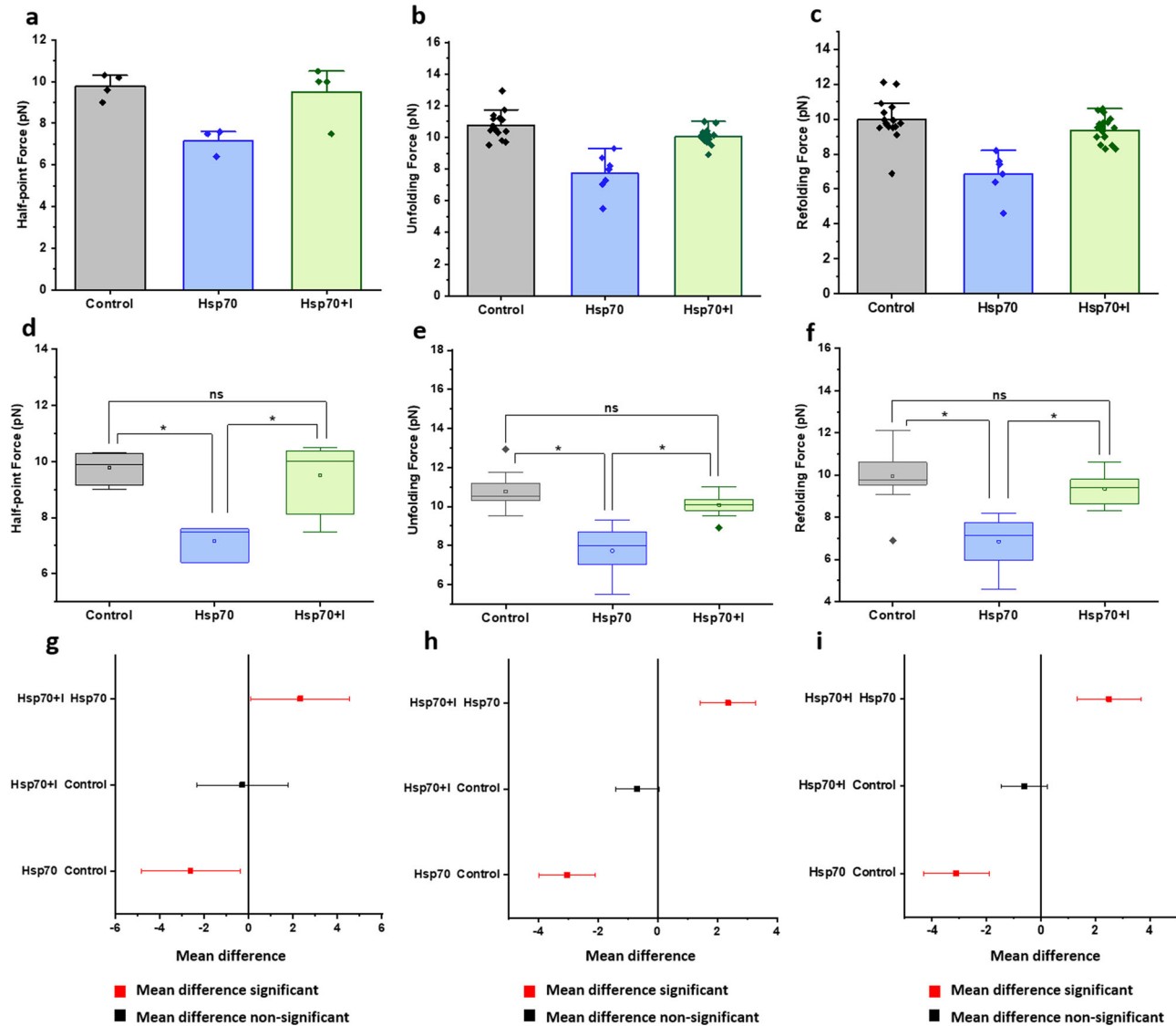

**Fig. 7 PFT-μ mediated Hsp70 inhibition restores talin mechanics. a** Half-point force measurement. In the presence of Hsp70, the half-point force of talin is 7.2 ± 0.4 pN, however, PFT-μ binding to Hsp70 relieves its unfoldase activity, upshifting the talin half-point force to 9.5 ± 0.7 pN. **b, c** We have also performed unfolding and refolding force-ramp experiments with Hsp70 both in the absence and presence of PFT-μ. We found that both unfolding and refolding forces increase in the presence of Hsp70●PFT-μ complex than only Hsp70, and overlap with those of control. ANOVA analysis has been performed for each of the **d, g** half-point forces; **e, h** unfolding force; **f, i** refolding force. We performed one-way ANOVA analysis to cross-check the statistical significance of different force data sets and corrected their mean differences by Bonferroni post hoc test at *$p \leq 0.05$. It has been observed that half-point force as well as unfolding/refolding forces with Hsp70 both in the presence and absence of PFT-μ are statistically significant, which is also evident from the mean comparison of their data sets. Since PFT-μ reverts the Hsp70 activity and thereby, talin exhibits innate mechanical stability as compared to control data, mean force differences are statistically non-significant between the Hsp70●PFT-μ complex and control. ($R^2 = 0.643$ for half-point force; 0.651 for unfolding force; and 0.522 for refolding force). For the half-point force analysis, $n = 4$ molecules are measured and averaged for each data set. For force analysis, unfolding forces are measured and averaged; $n = 15$ (control), $n = 7$ (Hsp70), $n = 17$ (Hsp70+inhibitor). For refolding force, $n = 16$ (control), $n = 6$ (Hsp70), $n = 20$ (Hsp70+inhibitor).

decreasing the unfolding free energy barrier ($\Delta G^{\ddagger}_{U}$) from 26.8 ± 0.8 to 21.7 ± 0.3 $k_BT$ (decreased by ~5 $k_BT$), inclining the mechanical free energy landscape of R3-IVVI towards the unfolded state (Table 1), which can also be explained from the downshifted folding dynamics. However, DnaKJE complex as a well-known foldase[52–57], has only been observed to restore the intrinsic folding ability of talin domain as comparable to the absence of any chaperones. Interestingly, DsbA as a foldase chaperone shifts the free energy landscape towards the folded state by increasing $\Delta G^{\ddagger}_{U}$ to 33.5 ± 0.5 $k_BT$ (increased by ~7 $k_BT$) (Table 1) which in turn increases the force range of folding dynamics. We assumed $A = 10^6 \, \text{s}^{-1}$ in Bell-like equation, which

gives an average unfolding barrier height within a range of 20–30 $k_BT$. Though $A$ has been reported to vary within a wide range of $10^6$ to $10^{13} \, \text{s}^{-1}$, a value of $10^6 \, \text{s}^{-1}$ is assumed to be practically relevant for attempt frequency based on the polymer scaling law and diffusion theory[58,59]. Attempt frequency is primarily changed due to the type of the reactive system such as proteins and reaction occurrence in solution, where it has been observed to decrease significantly than predicted by transition state theory[60]. In force spectroscopy approaches, the kinetics data are estimated either by Monte-Carlo simulations or by fitting to a different mathematical framework such as Bell-like, Cusp, and linear-cubic model which affects the attempt frequency

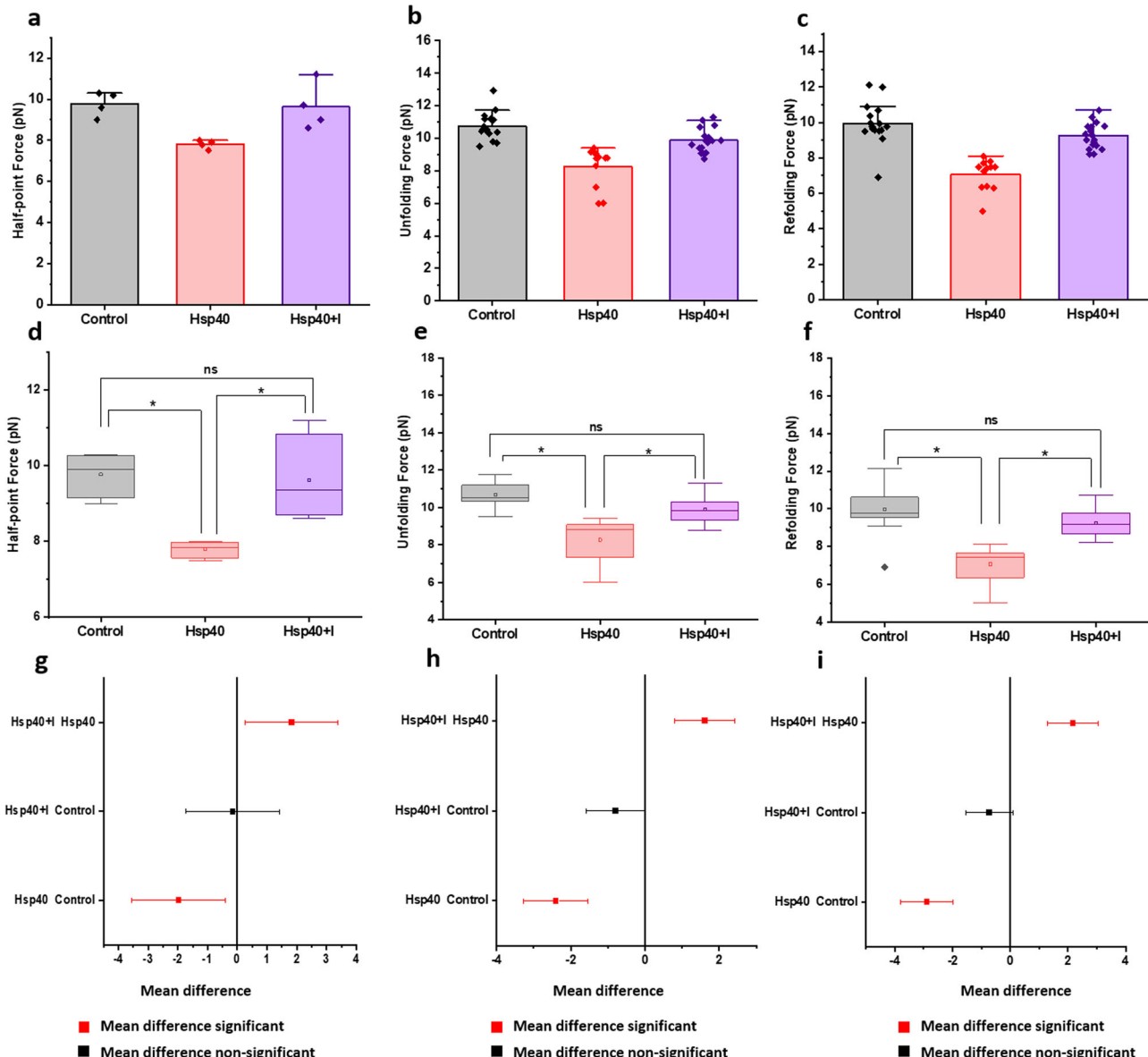

**Fig. 8 Talin mechanics with Hsp40 inhibition by KNK437.** Inhibition of Hsp40 chaperone promote innate talin mechanics by restoring **a** half-point force, **b** unfolding force, and **c** refolding force. ANOVA analysis has been performed for each of the **d**, **g** half-point force; **e**, **h** unfolding force; **f**, **i** refolding force. To cross-check the statistical significance of different force data sets, we performed one-way ANOVA analysis and observed that the force data with Hsp40 both in the presence and absence of its inhibitor are statistically significant, which is also evident from a mean comparison of their data sets. Since the inhibitor reverts the Hsp40 activity and thereby, talin exhibits innate mechanical stability as comparable to control data, mean force differences are statistically non-significant between the Hsp40 inhibitor complex and control. ($R^2 = 0.652$ for half-point force; 0.553 for unfolding force; and 0.608 for refolding force). For the half-point force analysis, $n = 4$ molecules are measured and averaged for each data sets. For force analysis, unfolding forces are measured and averaged; $n = 15$ (control), $n = 12$ (Hsp40), $n = 18$ (Hsp40+inhibitor). For refolding force, $n = 16$ (control), $n = 12$ (Hsp40), $n = 18$ (Hsp40+inhibitor).

value[27,61,62]. It has been estimated that the *A* value is higher ($10^6 \, \text{s}^{-1}$) in the Bell-like equation than both the cusp and linear-cubic model ($10^2 \, \text{s}^{-1}$) due to transition state dependency while calculating the reaction rates[27]. Our obtained range of barrier height is in well agreement to previously reported value of barrier height by different groups[63,64]. However, the *A* value is likely to differ from the values estimated by other force spectroscopic studies. This is due to the different reaction co-ordinates in force spectroscopic approaches, where the change in protein extension is probed with the experimental timescale and different folding mechanisms underlying the mechanical unfolding process[65]. Notably, these barrier height values also follow a similar changing

pattern with the energy values, obtained by equilibrium energy measurements: barrier height decreases in the presence of unfoldase and increases with the DsbA as foldase.

Furthermore, the changes in the unfolding barrier height values are in well agreement with previous single-molecule studies of protein folding dynamics. Brujić et al.[66] demonstrated the unfolding dynamics of ubiquitin by single-molecule force spectroscopy and observed that average unfolding barrier height of ubiquitin lies within $5\text{–}10 \, \text{k}_\text{B}\text{T}$ range. More recently, an optical tweezers study revealed that zippering of different domains in SNARE complex met only few $\text{k}_\text{B}\text{T}$. For example, the energy barrier of NTD and MD domain zippering were found to be 1

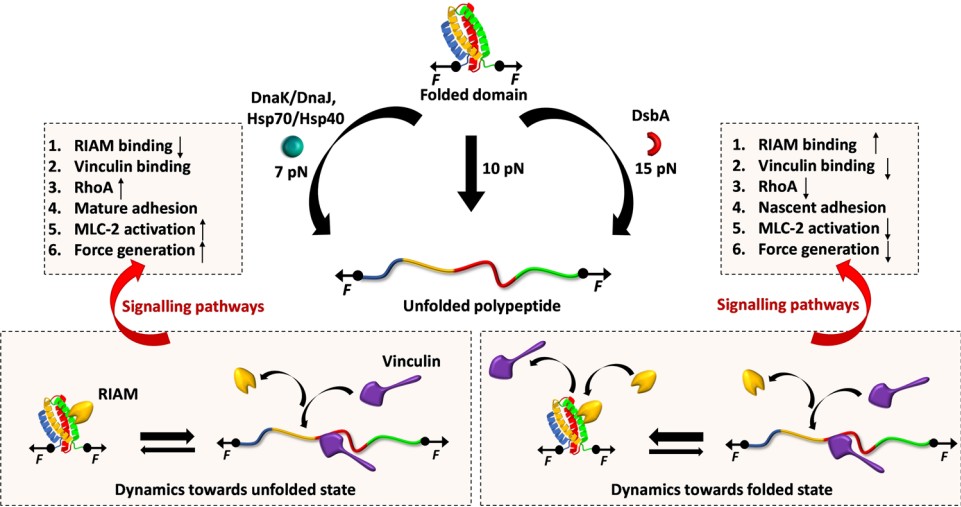

**Fig. 9 Plausible in vivo effect of Chaperone-modulated mechanical stability of talin.** Talin rod domains exhibit strong tension-dependent conformational changes which can further be modulated by both classes of chaperones: unfoldase and foldase. In the absence of chaperones, R3 unfolds at 10 pN force while it shifts to 15 pN or 7 pN, in the presence of foldase and unfoldase chaperones, respectively. This altered-mechanical stability could influence cellular physiology by tuning the downstream signaling pathways. Unfoldases such as Hsp70 and Hsp40 shift the folding dynamics of R3 domain towards the unfolded state, thereby, favoring the RIAM dissociation and vinculin interaction. Subsequently, this induces RhoA signaling, downstream MLC-2 activation, and thus, enhances the force generation at growing adhesion to facilitate cell migration. By contrast, foldase (DsbA) shifts the folding dynamics towards the folded state which subsequently promotes RIAM binding to the talin domains and localization in nascent adhesion. This negatively regulates the RhoA pathway which in turn abrogates MLC-2 activation, force generation, and cell migration.

and 2 $k_BT^{67}$. In addition, a single-molecule FRET study showed that the free energy barrier of CspTm folding lies within 4–11 $k_BT^{25}$. The barrier height in PrP protein has also been observed to be $2 \pm 1.2$ $k_BT$ ($5 \pm 3$ kJ/mol) from the unfolded state[68]. Importantly, these small changes in the free energy barrier of $\geq 3$ $k_BT$ could describe a single effective free energy barrier between the folded and unfolded states of protein and thus, signifies the two-state model of protein folding[69]. This chaperone-mediated reshaping of the free energy landscape apparently affects the mechanical stability of the R3 domain, allowing it to perform as a regulatory mechanical band-pass filter within a narrow force-frequency, to convert complex mechanical signals. Therefore, from a broader viewpoint, this could largely impact the force transduction into biochemical signaling by changing the force-dependent folding dynamics.

Mechanotransduction events through the FAs are largely dependent on robust and precise mechanical response of molecular force springs including talin, vinculin etc., which decode the mechanical signals by both the force magnitude and their temporal resolution. Talin rod domains have the exquisite ability to interpret these signals by force-driven structural dynamicity and concurrent modulation of complex interactome profiles. These rod domains have graded mechanical stability and unfold sequentially at increasing tension to control the adhesion dynamics[8,20]. Among them, R3 domain acts as the initial mechanosensor, initiating talin activation and subsequent signaling pathways. At low force (<5 pN), R3 domain engages to RIAM via folded state and becomes protected against mechanical stretching. The tension-induced unfolding at ~5 pN and concurrent RIAM dissociation is an essential criterion for vinculin binding to cryptic R3 helices (vinculin binding site, VBS). Recently, by pressure-dependent chemical shift, it has also been observed that R3 domain is thermodynamically tuned for interacting with either RIAM or vinculin which undergoes through a stepwise structural transition[2]. This mutual exclusive interaction of RIAM and vinculin to R3 finely calibrates downstream signaling cascades, reflecting the changes in cell morphology[70]. RIAM recruitment to talin-integrin mechanical linkage results in

filopodial protrusion in nascent adhesion that is eventually replaced by vinculin in mature adhesion[9]. The adhesion reinforcement results from the actomyosin contractility, which regulates the FA dynamics by a tension-dependent binding constant of antagonistic partners. For example, RIAM, DLC1 binding to folded domains, can stabilize talin by increasing the mechanical force threshold for vinculin binding. R3 stabilization has been reported to affect fibroblast matrix rigidity sensing and YAP (yes-associated protein) signaling, while destabilized R3 exhibits conformational dynamics and is more prone to tension-induced unfolding[71]. This unfolding at relatively low force, and subsequent vinculin binding plausibly activate downstream signaling pathways that decrease traction force generation. In mature adhesion, actomyosin contractility perturbs RIAM binding, abrogating its negative effect on RhoA signaling which enhances the tension at growing adhesion. However, as the adhesion matures and engages more proteins to the complex, the resultant force is decreased on individual talin molecules due to multiple parallel linkages. Recently, this force reduction on each mechanical linkage has been observed by FRET tension sensor and reported to decrease in mature FA complex compared to nascent adhesion at the cell periphery[72,73]. This means R3 domain again switches to RIAM binding state and reduces the force transmission through successive domains and thus, regulates FA dynamics. Interestingly, integrin-linked kinase, a protein in FA, has also been thought to be controlled by the active participation of the Hsp90 chaperones where they could modulate the folding–unfolding transitions of substrate proteins by changing their stability and therefore, their interaction with other partners[74]. Our study, from a larger viewpoint, suggests the plausible roles of chaperones in modulating the folding dynamics of adhesion proteins under force. This might play a pivotal role in mechanotransduction where force triggers conformational changes of mechanosensing proteins, thereby, regulating downstream signaling pathways (Fig. 9).

There is emerging evidence to suggest that molecular chaperones could accelerate the folding–unfolding transition of cytoskeletal proteins such as actin and tubulin[13]. Studies have shown

that chaperones like $\alpha\beta$ crystallin or calreticulin have played crucial roles in deciding mechanical properties of cells, like the resilience of cell adhesion, cell spreading, by regulating the expression of key FA proteins[13,14,18]. Small heat shock proteins have also been observed to regulate chronic biomechanical stress in heart tissue by regulating dynamics of key FA proteins like actin and filamin C[17]. In addition, tension-induced ubiquitylation of filamin A promotes chaperone-assisted autophagy which in turn dictates the roles of chaperones in FA assembly[75,76]. However, the underlying energy landscape of these chaperone-protein interactions, during mechanotransduction, has not been reported yet. Interestingly, we observed that Hsp70 and Hsp40, the eukaryotic homologs of DnaK and DnaJ, act as unfoldase by decreasing the unfolding barrier height from $26.8 \pm 0.8$ $k_BT$ to $23.1$ $k_BT$ (Fig. 4, Table 1), stabilizing its unfolded state. This shifting towards the unfolded state is also evident from the lower unfolding force in the presence of Hsp70/Hsp40 chaperones (Fig. 5a). During interaction with Hsp70 and Hsp40 in the cytosol, talin folding dynamics shifts to a lower force range (Supplementary Fig. 15), where it is mechanically poised to interact with vinculin or any other unfolded state interactors. Plausibly, in the case of higher Hsp70 expressions, talin-vinculin could predominantly interact, perturbing normal force-dependent processes in cells. By contrast, tunnel-associated chaperones possess the ability to increase the mechanical stability of their substrates. Such as oxidoreductase DsbA, a well-known model mechanical foldase, has the ability to increase the folding rate and mechanical stability of R3[24,77,78]. Hence, our results demonstrate that chaperones could modulate the mechanical stability of the FA protein talin, which is characterized by the change in the unfolding free energy barrier. From a broader perspective, our result may have a generic mechanistic insight of how talin response, and so their binding kinetics with other interactors, could be affected by chaperone interactions. For example, foldase chaperones facilitate the RIAM binding by shifting the energy barrier towards folded state and slowing the stepwise unfolding of the domain, whereas unfoldase chaperones promote vinculin binding by decreasing the unfolding free energy barrier. This could convert the switch response of R3 domain into a domain-locked response, disrupting the natural downstream signaling pathway to control the diverse cellular processes.

## Materials and methods

**Expression and purification of talin R3-IVVI**. We used the mechanically stable talin R3-IVVI domain as a model substrate, as previously reported in force spectroscopic studies[21,22]. For purification, the protein construct was transformed into the E. coli BL21 (DE3 Invitrogen) competent cells. Cells were grown in Luria broth at 37 °C with carbenicillin, till the O.D. becomes 0.6–0.8 at 600 nm. The cultures were then induced with 1 mM Isopropyl $\beta$-D-thiogalactopyranoside (IPTG, Sigma Aldrich) overnight at 25 °C. The cells were pelleted and re-suspended in 50 mM sodium phosphate buffer pH 7.4, containing 300 mM NaCl and 10% glycerol. Phenylmethylsulfonyl (PMSF) was used as a protease inhibitor followed by lysozyme for membrane lysis. After incubating the solution for 20 min at 4 °C, the dissolved pellet was treated with Triton-X 100 (Sigma Aldrich), DNase, RNase (Invitrogen), and 10 mM MgCl$_2$ and kept at 4 °C in the rocking platform. The cells were disrupted in French press and cell lysate was centrifuged at 11,000 rpm for 1 h. The protein was purified from the lysate using Ni$^{2+}$-NTA column of ÄKTA Pure (GE healthcare). For in vitro biotinylation of the polyprotein, and Avidity biotinylation kit was used and the biotinylated polypeptide was purified by Superdex-200 increase 10/300 GL gel filtration column in presence of Na-P buffer with 150 mM NaCl[78].

**Expression and purification of Chaperones (DnaJ, DnaK, GrpE, and DsbA)**. For purification of DnaJ and GrpE protein purification, these constructs were transformed into E. coli (DE3) cells and grown in Luria broth media at 37 °C with respective antibiotics (carbenicillin and ampicillin, 50 μg/ml). The cells were induced overnight with IPTG (Sigma Aldrich) at 25 °C. Cells were centrifuged and the pellet was re-suspended in 50 mM sodium phosphate buffer, 10% glycerol, and 300 mM NaCl, pH 7.4. The solution was then incubated with PMSF and lysozyme.

DNase, RNase, MgCl$_2$, and Triton-X 100 were mixed subsequently at 4 °C. Then the cells were disrupted in French press and the lysate was extracted and purified using Ni$^{2+}$-NTA affinity column of ÄKTA Pure (GE healthcare). We used 20 mM imidazole containing buffer as a binding buffer and 250 mM imidazole containing sodium phosphate buffer as elution buffer. We purified full-length DnaK chaperone from the cell lysate using the Ni$^{2+}$-NTA column, pre-loaded with Mge1[79]. Then the DnaK protein was eluted with buffer containing 2 mM ATP[79]. For DsbA, the pellet was dissolved in 50 mM Tris buffer with 1 mM EDTA and 20% (w/v) sucrose, pH 8.0. After 20 min of ice incubation, the mixture was provided with DNase, RNase, and PMSF. Then centrifuged the solution followed by collecting both the supernatant (S-1) and pellet. The pellet was dissolved in 20 mM Tris solution and again centrifuged and the supernatant (S-2) was collected. The purification was done by purifying these two supernatants with 1 M NaCl containing Tris buffer using a Hi-Trap Q-FF anion exchange column of ÄKTA Pure (GE healthcare). To maintain the oxidation activity of DsbA 0.3% H$_2$O$_2$ was added to the purified solution and kept overnight. The protein was further purified by size exclusion chromatography using Superdex-200 increase 10/300 GL gel filtration column in presence of 150 mM NaCl (Supplementary Fig. 16)[24]. We purchased Hsp70 and Hsp40 chaperones from ORIGENE Technologies. AR03018PU-N (Hsp70, human recombinant protein) and AR09036PU-L (Hsp40, human recombinant protein). The purity of Hsp70 and Hsp40 is >90% and ≥95% as determined by SDS-PAGE analysis.

**Preparation of glass slide and coverslips**. During the magnetic tweezers experiment, the glass slides were washed with Hellmanex III (1.5%) solution (Sigma Aldrich) followed by washing with double-distilled water. It was then soaked in a mixture containing concentrated hydrochloric acid (HCl) and methanol (CH$_3$OH). The slides were then treated with concentrated sulfuric acid (H$_2$SO$_4$) followed by washing in double-distilled water. The glass slides were put into gently boiling water and dried. To activate the glass surface, the glass slides were dissolved in the ethanol solution of 1% (3-Aminopropyl) trimethoxysilane (Sigma Aldrich, 281778) for 15 min. Then the glasses were washed with ethanol for removing the unreacted silane and baked at 65 °C. The coverslips were washed with Hellmanex III (1.5%) solution for 15 min followed by ethanol and dried in the oven for 10 min. After sandwiching the glass and coverslips, the chamber was filled with glutaraldehyde (Sigma Aldrich) for an hour. Then the chambers were flushed with reference beads (2.5–2.9 μm, Spherotech, AP-25-10) and HaloTag (O4) ligand (Promega, P6741). To avoid non-specific interaction the glass chambers were washed with blocking buffer (20 mM Tris-HCl, 150 mM NaCl, 2 mM MgCl$_2$, 0.03% NaN$_3$, 1% bovine serum albumin, pH 7.4) for 5 h at room temperature[19,80].

**Magnetic tweezers experiment**. Magnetic tweezers set-up was built on an inverted microscope using ×63 oil-immersion objective that is attached with a nanofocusing piezo actuator. A linear voice- coil is located above the sample chamber to control the position of the magnet. Images were acquired using ximea camera (MQ013MG-ON). Details information regarding the bead tracking, image processing, and force calibration are described previously[23]. We have also provided a detailed description of the force calibration method in the supplementary information (Supplementary Fig. 17). Magnetic tweezers experiment was performed with 1–10 nM of protein in a buffer containing 1× PBS buffer (pH 7.2) and 10 mM ascorbic acid[81,82]. Ascorbic acid was used as an antioxidant for preventing the oxidative damage of the protein. After passing the protein through the chamber, streptavidin-coated paramagnetic beads (Dynabeads M-270, cat. No. 65305) were passed through the chamber, where they get adhered to the biotinylated Avi-Tagged protein. Folding and unfolding dynamics of the protein were observed by applying different force-ramp and force-clamp protocols. For experiments with chaperones, we used 0.5 μM, 1 μM, and 3 μM of DnaJ, whereas DnaK was introduced at 1.5 μM, 3 μM, and 10 μM concentration. With DnaKJ and DnaKJE complex, we used 1 μM DnaJ and 3 μM DnaK with 5 μM GrpE. During experiments with DnaK, DnaKJ, and DnaKJE, the buffers were supplemented with 10 mM ATP and 10 mM MgCl$_2$. The buffers were exchanged every 30 min with fresh ATP to provide sufficient ATP supply.

**Statistics and reproducibility**. We have performed the statistical analysis by one-way ANOVA, followed by Bonferroni post hoc test for the pair comparisons of the data sets. All the data are reproduced and detailed information regarding statistics and reproducibility are mentioned in the figure legends.

**Reporting summary**. Further information on research design is available in the Nature Research Reporting Summary linked to this article.

## Data availability
The source data underlying the graphs are available as Supplementary Data 1.

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

## Acknowledgements

We thank Ashoka University for support and funding. S.H. thanks DBT Rama-lingaswami Fellowship and DST SERB Core Research Grant for funding. We thank Dr. Koyeli Mapa of Shiv Nadar University for kindly sharing with us the clones of DnaK, DnaJ, GrpE proteins. We sincerely acknowledge professor Julio Fernandez (Columbia University) for helping us with the magnetic tweezers set-up. We thank professor LS Shashidhara (Ashoka University), and Dr. Edward C. Eckels (Columbia University Medical Center, New York) for the critical analysis and discussion of this work.

## Author contributions

S.H., S.C. designed the project and wrote the manuscript. D.C., S.B., M.B., S.C. did the experiments and analyzed the data.

## Competing interests

The authors declare no competing interests.
