## [Peer Review File · Communications Biology]

Reviewers' comments:

Reviewer #1 (Remarks to the Author):

The manuscript "Real time observation of chaperone-modulated talin mechanics under single molecule resolution" by Chakraborty presents the very interesting and novel hypothesis that chaperones might be involved in the regulation of talin folding/unfolding under mechanical force which was addressed by force spectroscopic measurements.

These force spectroscopic measurements and the analyses have been done thoroughly and the principal results clearly show that DnaK, DnaJ, or mammalian homologues Hsp70 and Hsp40, reduce the unfolding force of the talin R3-IVVI domain, whereas DsbA increases its stability and thus the unfolding force.

In particular, the results obtained for Hsp70 (the mammalian homologue) are very interesting and relevant with respect to mechanotransduction, as this protein has been found to be one of three chaperones (HSPA2/Hsp70, HSPB1/Hsp27 and CALR) in the consensus integrin adhesome dissected by Horton et al. (2015, NCB, doi: 10.1038/ncb3257). This work should be mentioned and cited.

The authors should furthermore extend the experiments regarding Hsp70 and Hsp40 with suitable inhibitors against these proteins, to see whether inhibitor treatments reverse the chaperone-induced effect on the talin domain.

In Fig. 7 the authors present a model of the "Plausible in vivo effect of Chaperone-modulated mechanical stability of talin". This model comprises many potential impacts of the chaperone activity on focal adhesion dynamics and mechanotransductive processes. This speculation is highly interesting, but this work needs a first step into the validation of this model in cells. A promising experiment to be done, would be to analyse what happens to focal adhesions (e.g., staining against vinculin) and the actin cytoskeleton e.g. after inhibition with an inhibitor against Hsp70. The experiments could be done in cells that make pronounced focal adhesions, such as fibroblasts (which were e.g. also used in the abovementioned Horton et al. paper). It should be quantified, e.g. from confocal images, whether/how Hsp70 inhibitor treatments affect the focal adhesion dimensions and actin cytoskeletal organisation (e.g., stress fibre formation).

Maybe I missed it, but I didn't find information about where the Hsp70 and Hsp40 are coming from. Were they expressed and purified in the same way as the talin construct DnaK, DsbA, GrpE?

In conclusion, the manuscript demonstrates data that shows the potential of chaperones to influence talin mechanics. If that would be the case, this would open up a whole new level of regulation of mechanotransductive processes. The data is thus relevant. However, this potential and intriguing hypothesis should additionally be addressed with first experiments, as proposed, in cells to strengthen the claims and the model.

Minor remarks:

- The sentence "Since Hsp70/Hsp40 are the ubiquitous cytoplasmic chaperones, it interacts with different cytosolic proteins and talin being a critical mechanosensitive cytosolic protein, possess a high probability to interact with the chaperone complex." Sounds grammatically wrong to me.
- There is an "in" missing in the sentence "..., which is also observed" in "the force-decrease scan as sudden decrease in the extension at particular peak forces".
- Must be "if" instead of "in" in this sentence: "The unfolding force of talin has been observed to decrease with the unfoldase chaperone, in compared to the"
- An "are" is missing in the sentence: "...eukaryotic cytosol, they" are "most likely to interact with talin and thus, could change the talin folding dynamics by modulating their mechanical stability."
- Must be "have" in the sentence: "Additionally, Hsp70 and Hsp40 has been observed to significantly shift..."
- Must be "a" not "the": "...complexes and observed the similar pattern in the unfolding force-higher unfolding force in the presence..."
- A "the" is missing in the sentence: "..., we observed the unfolding force is" the "same as in the absence of any chaperones"

- Must be "changes" (lowercase): "These Changes in the unfolding barrier height values".
- A "the" is missing: "Talin rod domains have" the "exquisite ability to interpret"
- The sentence sounds grammatically wrong: "to be decreased in mature focal adhesion complex than nascent adhesion at cell periphery." Maybe better "compared to" than "than".
- Are there references for this? The paper by Radovanac et al (2013, EMBO J, 10.1038/emboj.2013.90 should be mentioned and discussed here: "Interestingly, focal adhesion has also been thought to be controlled by active participation of the molecular chaperones with different adhesion proteins, where they could modulate the folding-unfolding transitions of substrate proteins by changing their mechanical stability and therefore, their interaction with other partners."
- Must be "have", "roles", "the cell". Furthermore, it would be interesting to mention that calreticulin is among the chaperones that were found in the consensus integrin adhesome (see above, Horton et al. paper): "Studies have shown that chaperones like $\alpha\beta$ crystallin or calreticulin has played crucial role in deciding mechanical properties of cell, like resilience of cell adhesion, cell spreading, by regulating expression of key focal adhesion proteins.
- Must be "get": "...were passed through the chamber, where they gets adhered to the biotinylated Avi-Tagged protein."
- Reference 23 has now been published in Chemical Science, please update.
- Figure should use different colour codes for the different chaperones (as was done e.g. in Suppl. Fig. 11). This might facilitate to follow the data in an easier manner and to distinguish better the different experimental conditions.
- In the Fig. 2 caption it must be "error bars".
- A few times "signaling" has been used, but most of the times "signalling", use one or the other spelling.
- The authors should show (in the Supplementary Information) Western blots of the purified proteins.

Reviewer #2 (Remarks to the Author):

Referee report for "Real time observation of chaperone-modulated talin mechanics under single molecule resolution" by Chakraborty et al.

The authors study the talin rod domain R3 (or a mutant thereof) using single-molecule magnetic tweezers in the absence and presence of several different chaperones. The authors find that some chaperones stabilize the protein and increase the force required for unfolding, while other destabilize it and reduce the force for unfolding.

Overall, I find the data of good quality and this is a useful characterization of this biologically relevant domain. However, I see some questions and issue that should be addressed to increase the clarity of the results and potentially broaden the relevance.

1) In several places, the authors claim to have "determined the fundamental mechanism of this altered mechanical stability" (Abstract) and say that "we explicitly demonstrate the underlying physical mechanism of this altered mechanical response" (Introduction).

However, the explanation provided is to quantify how the energy barrier has changed. While it is fine to report the energy barrier values, I don't really see this as more fundamental as reporting the force and the distance to the transition state (after all, force x distance = energy). It leaves me wondering what is happening at a molecular level. What kind of interactions destabilize or stabilizes the talin fold? I would phrase this more carefully and, if possible, try to at least speculate about a microscopic and mechanistic explanation.

2) This might not be completely critical, but it would be useful to perform at least limited test measurements on whether the observed trends also hold up with the wild-type R3 domain, in addition to the IVVI mutant used.

3) The key results of the paper are a number of unfolding/refolding force values. These appear to be reported as means +/- standard error of the mean? Is this what was done? It should be

explicitly stated (e.g. on page, when the first values are reported in the form $x \pm y$) how the values were generated. Importantly, rupture forces at a given loading rate follow a Bell-Evans distribution, not a Gaussian distribution. This should be considered when analyzing the data.

4) The author state "Protein exhibits force-independent unfolding transition due to the rigid folded state, and thus, the unfolding kinetics fits better to Bell-like equation."

This is unclear/incorrect for several reasons

--- Folding is certainly NOT force-independent! The authors show themselves how for their system the protein unfolds under force, with a rate that increases with force (Figure 4).

--- The Bell equation is precisely used to quantify the force dependence!

--- The statement "fits better" is unclear – better than what?

5) I think it is good that the authors, in addition to studying prokaryotic chaperones that would not likely interact with talin, also eukaryotic chaperones Hsp70 and Hsp40. Still their statement "Since these chaperones reside in eukaryotic cytosol, they most likely to interact with talin" is an overstatement. Many things are in the cytosol and this does not mean they all interact!

6) How were the free energy differences computed? From the fits of the rates by Equations 2 and 3? Doesn't the "molecular attempt frequency" A influence the results? How well is this known? Do the results match the free energy differences suggested by the equilibrium measurements via $\text{Energy} = [\text{midpoint force}] \times [\text{distance between folded and unfolded state}]$?

7) I am not sure I understand the point the authors are trying to make in the Discussion (page 9), when they compare their energies to the literature. The authors find folding free energies in the range of 20-30 kT, and changes by the chaperones of a few kT. However, they then go on to cite a number of references where the folding free energy was on the order of a few kT.

I do not doubt that the results presenting in the paper are in a plausible range of values, but wouldn't more relevant references be cases where the actual folding free energy is on the order of 20-30 kT and then external factors change that number by a few kT? One example also using magnetic tweezers is e.g.

Multiplexed protein force spectroscopy reveals equilibrium protein folding dynamics and the low-force response of von Willebrand factor.

Löf A, Walker PU, Sedlak SM, Gruber S, Obser T, Brehm MA, Benoit M, Lipfert J. Proc Natl Acad Sci U S A. 2019 Sep 17;116(38):18798-18807. doi: 10.1073/pnas.1901794116.

where

$[\text{midpoint force}] \times [\text{distance between folded and unfolded state}]$ appear to be in the range of 20-30 kT, which changes of a few kT due to conditions.

8) Given the many pairwise comparisons of forces in the manuscript, it would be good to compute statistical significances for the differences and to also report them in the figures (e.g. in Figure 2).

Minor points:

- I would probably be good to write out/define "RIAM" in the Abstract.

- In the Abstract, it would help to state also the midpoint force in the absence of chaperones. It is stated that "holdase chaperones reduce their unfolding force to ~ 6 pN", but one wonders what the force was before this intervention.

- The authors emphasize the "real-time" nature of their measurements, e.g. in the title of the paper and twice in one sentence on page 4. Is this really so relevant? Having time resolution is important, but it seems to me that recording the time traces and post-processing them would provide exactly the same information as real-time processing?

- "Equation" or "Eq." should be capitalize when referring to the specific equation.

- Something is missing in "Magnetic tweezers technology, using fore-clamp protocol, allows us".

Reviewers' comments:

Reviewer #1 (Remarks to the Author):

The manuscript "Real time observation of chaperone-modulated talin mechanics under single molecule resolution" by Chakraborty presents the very interesting and novel hypothesis that chaperones might be involved in the regulation of talin folding/unfolding under mechanical force which was addressed by force spectroscopic measurements.

These force spectroscopic measurements and the analyses have been done thoroughly and the principal results clearly show that DnaK, DnaJ, or mammalian homologues Hsp70 and Hsp40, reduce the unfolding force of the talin R3-IVVI domain, whereas DsbA increases its stability and thus the unfolding force.

In particular, the results obtained for Hsp70 (the mammalian homologue) are very interesting and relevant with respect to mechanotransduction, as this protein has been found to be one of three chaperones (HSPA2/Hsp70, HSPB1/Hsp27 and CALR) in the consensus integrin adhesome dissected by Horton et al. (2015, NCB, doi: 10.1038/ncb3257). This work should be mentioned and cited.

We appreciate reviewer#1 for the positive comments on our manuscript. These comments are really helpful to improve the quality of the manuscript. We would also like to thank reviewer#1 for suggesting us the reference. We have included this reference in the revised manuscript (reference number 18).

The authors should furthermore extend the experiments regarding Hsp70 and Hsp40 with suitable inhibitors against these proteins, to see whether inhibitor treatments reverse the chaperone-induced effect on the talin domain.

We thank reviewer#1 for the valuable comment on the experiments with Hsp70 and Hsp40 in the presence of their inhibitors. We have performed the experiments with the Hsp chaperones in the presence of their inhibitors and observed that inhibitor treatments reverse the chaperone-induced effect on the talin.

We have performed the experiments with Hsp70 in the presence of its inhibitor pifithrin- μ (PFT- μ), which is known to bind the substrate-binding domain of Hsp70 and block its activity (Leu et al., Mol Cell., 2009; Balaburski et al., Mol Cancer Res., 2013). We observed that PFT- μ binding reverts the Hsp70-induced effect on the talin domain. In the presence of Hsp70, the half-point force of talin is 7.2 pN, however, PFT- μ binding to Hsp70 relieves its unfoldase activity, upshifting the talin half-point force to 9.5 pN (Figure 1A). To reconcile the Hsp70-induced effect on talin mechanics, we have also performed the force-ramp experiments with Hsp70 both in the absence and presence of PFT- μ . We found that both unfolding and refolding forces increase in the presence of Hsp70•PFT- μ complex than only Hsp70, and overlaps with those of control (Figure 1B and 1C). Similarly, we performed the experiments with Hsp40 in the presence of KNK437 as its inhibitor (Yokota et al., Cancer Res., 2000; Kumar et al., JBC. 2005) and observed that Hsp40-mediated change in talin half-point force is also relieved upon KNK437 treatment (Figure 2A). This reversion of Hsp40 effect have also been observed in force-ramp experiment with Hsp40•KNK437 complex, where both the unfolding and refolding forces are higher, if compared to only in the presence of the Hsp40 chaperone (Figure 2B and 2C).

Additionally, we have performed one-way ANOVA to crosscheck the statistical significance of the force data sets with Hsp chaperones and corrected their mean differences by Bonferroni

post hoc test at $*p \leq 0.05$. It has been observed that half-point force as well as unfolding/refolding force with Hsp70, both in the presence and absence of pifithrin- μ are statistically significant, which is also evident from mean comparison of their data sets (Figure 1D to 1I). Since PFT- μ reverts the Hsp70 activity and thereby, talin exhibits innate mechanical stability as comparable to control data, mean force differences are statistically non-significant between the Hsp70•PFT- μ complex and control. Similarly, we observed that the force range with Hsp40-inhibitor complex possess no significant difference with that of control data, indicating that Hsp40 inhibition restores normal talin mechanics (Figure 2D to 2I). These experiments strengthen our observation that change in talin mechanical stability certainly arise from the Hsp70/Hsp40 interactions and thereby, reconcile the hypothesis of the chaperone-altered mechanical stability in talin. For the convenience of the reviewer, we have provided the figure below and have also included this section in the revised manuscript.

Figure 1: PFT- μ mediated Hsp70 inhibition restores talin mechanics: (A) Half-point force: In the presence of Hsp70, the half-point force of talin is 7.2 ± 0.4 pN, however, PFT- μ binding to Hsp70 relieves its unfoldase activity, upshifting the talin half-point force to 9.5 ± 0.7 pN. **(B and C) Unfolding and refolding force:** We have also performed the force-ramp experiments with Hsp70 both in the absence and presence of PFT- μ . We found that both unfolding and refolding forces increase in the presence of Hsp70•PFT- μ complex than only Hsp70, and overlaps with those of control. **(D-I) ANOVA analysis:** We performed one-way ANOVA analysis to crosscheck the statistical significance of different force data sets and corrected their mean differences by Bonferroni post hoc test at $*p \leq 0.05$. It has been observed that half-point force as well as unfolding/refolding forces with Hsp70 both in the presence and absence of PFT- μ are statistically significant, which is also evident from mean comparison of their data sets. Since PFT- μ reverts the Hsp70 activity and thereby, talin exhibits innate mechanical stability as comparable to control data, mean force differences are statistically non-significant between the Hsp70•PFT- μ complex and control. For the half-point force analysis, $n=4$ molecules are measured and averaged for each data sets. (R sq. = 0.643 for half-

point force; 0.651 for unfolding force; and 0.522 for refolding force). For the force analysis, unfolding forces are measured and averaged; $n=15$ (control), $n=7$ (Hsp70), $n=17$ (Hsp70+inhibitor). For refolding force; $n=16$ (control), $n=6$ (Hsp70), $n=20$ (Hsp70+inhibitor).

Figure 2: Talin mechanics with Hsp40 inhibition: Inhibition of Hsp40 chaperone promote innate talin mechanics by restoring (A) half-point force, (B) unfolding force, and (C) refolding force. (D-I) ANOVA analysis: To crosscheck the statistical significance of different force data sets, we performed one way ANOVA analysis and observed that the force data with Hsp40 both in the presence and absence of its inhibitor are statistically significant, which is also evident from mean comparison of their data sets. Since the inhibitor treatment reverts the Hsp40 activity and thereby, talin exhibits innate mechanical stability as comparable to control data, mean force differences are statistically non-significant between the Hsp40-inhibitor complex and control. ($R^2 = 0.652$ for half-point force;

0.553 for unfolding force; and 0.608 for refolding force). For the half-point force analysis, $n=4$ molecules are measured and averaged for each data sets. For the unfolding force analysis, unfolding forces are measured and averaged; $n=15$ (control), $n=12$ (Hsp40), $n=18$ (Hsp40+inhibitor). For refolding force, $n=16$ (control), $n=12$ (Hsp40), $n=18$ (Hsp40+inhibitor).

In Fig. 7 the authors present a model of the “Plausible in vivo effect of Chaperone-modulated mechanical stability of talin”. This model comprises many potential impacts of the chaperone activity on focal adhesion dynamics and mechanotransductive processes. This speculation is highly interesting, but this work needs a first step into the validation of this model in cells. A promising experiment to be done, would be to analyse what happens to focal adhesions (e.g., staining against vinculin) and the actin cytoskeleton e.g. after inhibition with an inhibitor against Hsp70. The experiments could be done in cells that make pronounced focal adhesions, such as fibroblasts (which were e.g. also used in the abovementioned Horton et al. paper). It should be quantified, e.g. from confocal images, whether/how Hsp70 inhibitor treatments affect the focal adhesion dimensions and actin cytoskeletal organisation (e.g., stress fibre formation).

We appreciate the comment of reviewer#1 regarding the Hsp inhibitor treatment in cultured fibroblast cells.

Talin is a mechanosensitive protein and thus, any change in its mechanical strength seems to affect its force-dependent interaction with other binding partners. Interestingly, we observed that chaperone modulates the mechanical stability of talin and thus, we hypothesized that chaperone-modulated mechanical stability of talin could have a “plausible in vivo effect”. From our observation, we have speculated that foldase chaperone could favour talin interaction with RIAM or actin which are folded state interactors, while unfoldase facilitates binding of unfolded state interactors such as vinculin.

As suggested by reviewer#1, fibroblast cells could be treated with the Hsp inhibitors to validate the mechanical effect of chaperones on focal adhesion (FA) proteins and thus, the adhesion dynamics. However, it is well known that focal adhesion is a multiprotein complex and these chaperones not only control the talin mechanics but also tune the stability of other important FA proteins such as actin, paxillin, and ILK, possessing non-specificity in modulating the folding of their FA substrates (Grantham et al., Front. Genet. 2020; Mao et al., J. Biol. Chem., 2004, Radovanac et al., 2013). Additionally, it have been recently reported that cAMP oscillations and PKC α proteolysis is dependent on the chaperone activity (Rinaldi et al., Nat. Commun. 2019). Thus, the inhibition of chaperone activity would exhibit diverse off-target effects in FA dynamics and growth, which could be multifactorial. We agree with reviewer#1 that an in vivo study could be interesting, but it cannot precisely conclude whether the changes are specifically because of the chaperone-regulated talin mechanics or any other focal adhesion proteins.

Therefore, we would like to keep it as a model and an open topic until a suitable experiment/instrumentation is built for this purpose. We used the ‘plausible’ term while illustrating the model for the same reason. We take the suggestion of reviewer#1 positively and will include this in our future objectives. While this being said, as of now we would be happy to remove figure 11 (old figure 7) from the manuscript if the reviewer suggests.

Maybe I missed it, but I didn’t find information about where the Hsp70 and Hsp40 are coming from. Were they expressed and purified in the same way as the talin construct DnaK, DsbA, GrpE?

We thank the reviewer for the comment. We have purchased Hsp70 and Hsp40 from ORIGENE Technologies. The catalog no are: AR03018PU-N (Hsp70, human recombinant protein) and AR09036PU-L (Hsp40, human recombinant protein). The purity of Hsp70 is >90% as

determined by SDS-PAGE analysis and Hsp40 is $\geq 95\%$ pure. We have also mentioned it in the “materials and method” section of the revised manuscript.

In conclusion, the manuscript demonstrates data that shows the potential of chaperones to influence talin mechanics. If that would be the case, this would open up a whole new level of regulation of mechanotransductive processes. The data is thus relevant. However, this potential and intriguing hypothesis should additionally be addressed with first experiments, as proposed, in cells to strengthen the claims and the model.

We thank reviewer#1 for the comment on the cell line experiment to strengthen the plausible model of chaperone-regulated mechanotransduction in focal adhesion. We have discussed it in detail on page 5.

Minor remarks:

- The sentence “Since Hsp70/Hsp40 are the ubiquitous cytoplasmic chaperones, it interacts with different cytosolic proteins and talin being a critical mechanosensitive cytosolic protein, possess a high probability to interact with the chaperone complex.” Sounds grammatically wrong to me.

We thank the reviewer for pointing this out and we have changed this in the revised manuscript.

- There is an “in” missing in the sentence “..., which is also observed” in “the force-decrease scan as sudden decrease in the extension at particular peak forces”.

We apologize to the reviewer for the mistake. We have corrected it in the revised manuscript.

- Must be “if” instead of “in” in this sentence: “The unfolding force of talin has been observed to decrease with the unfoldase chaperone, in compared to the”

We apologize to the reviewer for the mistake. We have corrected it in the revised manuscript.

- An “are” is missing in the sentence: “...eukaryotic cytosol, they” are “most likely to interact with talin and thus, could change the talin folding dynamics by modulating their mechanical stability.”

We thank reviewer#1 for the correction and we have corrected it in the revised manuscript.

- Must be “have” in the sentence: “Additionally, Hsp70 and Hsp40 has been observed to significantly shift...”

We thank the reviewer for pointing out the mistake and we have changed this in the revised manuscript.

- Must be “a” not “the”: “...complexes and observed the similar pattern in the unfolding force-higher unfolding force in the presence...”

We apologize to the reviewer for the mistake. We have corrected it in the revised manuscript.

- A “the” is missing in the sentence: “..., we observed the unfolding force is” the “same as in the absence of any chaperones”

We thank the reviewer for the correction. We have changed this in the revised manuscript.

- Must be “changes” (lowercase): “These Changes in the unfolding barrier height values”.

We thank the reviewer for pointing out the mistake and we have changed this in the revised manuscript.

- A “the” is missing: “Talin rod domains have” the “exquisite ability to interpret”

We thank the reviewer for the correction. We have changed this in the revised manuscript.

- The sentence sounds grammatically wrong: “to be decreased in mature focal adhesion complex than nascent adhesion at cell periphery.” Maybe better “compared to” than “than”.

We thank the reviewer for pointing this out and we have changed this in the revised manuscript.

- Are there references for this? The paper by Radovanac et al (2013, EMBO J, 10.1038/emboj.2013.90 should be mentioned and discussed here: “Interestingly, focal adhesion has also been thought to be controlled by active participation of the molecular chaperones with different adhesion proteins, where they could modulate the folding-unfolding transitions of substrate proteins by changing their mechanical stability and therefore, their interaction with other partners.”

We thank reviewer#1 for suggesting us the reference and we have cited this reference there in the revised manuscript.

- Must be “have”, “roles”, “the cell”. Furthermore, it would be interesting to mention that calreticulin is among the chaperones that were found in the consensus integrin adhesome (see above, Horton et al. paper): “Studies have shown that chaperones like $\alpha\beta$ crystallin or calreticulin has played crucial role in deciding mechanical properties of cell, like resilience of cell adhesion, cell spreading, by regulating expression of key focal adhesion proteins.

We thank the reviewer for suggesting us the changes and the reference. We have made the necessary changes in the manuscript and cited this reference there in the revised manuscript.

- Must be “get”: “...were passed through the chamber, where they gets adhered to the biotinylated Avi-Tagged protein.”

We apologize to the reviewer for the mistake. We have corrected it in the revised manuscript.

- Reference 23 has now been published in Chemical Science, please update.

We thank reviewer#1 for the comment and we have updated this citation in the revised manuscript.

- Figure should use different colour codes for the different chaperones (as was done e.g. in Suppl. Fig. 11). This might facilitate to follow the data in an easier manner and to distinguish better the different experimental conditions.

We thank the reviewer for the comment. We have changed the colour codes for different chaperones and included in the revised manuscript.

- In the Fig. 2 caption it must be “error bars”.

We thank the reviewer for pointing out the mistake and we have changed this in the revised manuscript.

- A few times “signaling” has been used, but most of the times “signalling”, use one or the other spelling.

We thank reviewer#1 for the comment. We have changed the spelling to “signaling” throughout the revised manuscript.

- The authors should show (in the Supplementary Information) Western blots of the purified proteins.

We thank reviewer#1 for the comment. We purified the protein by affinity and size exclusion chromatography, followed by the SDS-PAGE analysis to check the purity level of proteins.

Unfortunately, we don't have the western facility in our lab and due to the pandemic, accessing labs outside the institute is very restricted. Thus we have included the SDS-PAGE pictures in the revised manuscript. However, if the reviewer feels that these studies are absolutely necessary to satisfy the speculation of our work, we will surely perform them. For the convenience of the reviewer, we have provided the figures below.

Figure 3: SDS-PAGE of protein samples: We have performed the SDS-PAGE of the SEC (size exclusion chromatography) purified protein samples. **(A)** DsbA: We observed DsbA expression and its purification; **(B)** DnaK and DnaJ.

We appreciate reviewer#1 for the detailed comments on our manuscript. These comments are really helpful to improve the quality of the manuscript.

Reviewer #2 (Remarks to the Author):

Referee report for “Real time observation of chaperone-modulated talin mechanics under single molecule resolution” by Chakraborty et al.

The authors study the talin rod domain R3 (or a mutant thereof) using single-molecule magnetic tweezers in the absence and presence of several different chaperones. The authors find that some chaperones stabilize the protein and increase the force required for unfolding, while other destabilize it and reduce the force for unfolding.

Overall, I find the data of good quality and this is a useful characterization of this biologically relevant domain. However, I see some questions and issue that should be addressed to increase the clarity of the results and potentially broaden the relevance.

We would like to thank reviewer#2 for the constructive analysis of the manuscript, which has substantially improved the quality of the manuscript.

1) In several places, the authors claim to have “determined the fundamental mechanism of this altered mechanical stability” (Abstract) and say that “we explicitly demonstrate the underlying physical mechanism of this altered mechanical response” (Introduction).

However, the explanation provided is to quantify how the energy barrier has changed. While it is fine to report the energy barrier values, I don't really see this as more fundamental as reporting the force and the distance to the transition state (after all, force x distance = energy).

It leaves me wondering what is happening at a molecular level. What kind of interactions destabilize or stabilizes the talin fold? I would phrase this more carefully and, if possible, try to at least speculate about a microscopic and mechanistic explanation.

We apologize to the reviewer for the confusion in the manuscript. To avoid confusion, we have removed these statements and rephrased it in the revised manuscript.

We observed that the chaperones reshape the energy landscape of protein folding by changing the height of the unfolding energy barrier for the folded state, keeping their distance to the transition state constant. For example, unfoldase chaperone lowers the unfolding energy barrier by stabilizing the extended state of the substrate, which in turn affects the refolding rate of the substrate under force. This has also been confirmed from the mechanical chevron plot in the presence of different unfoldases, where the intersection force decreases with the unfoldase chaperones and increases with DsbA (foldase). For example, DnaJ downshift this force to 7.9 pN from 9.8 pN. However, the DnaKJE complex as a well-known foldase has only been observed to restore the intrinsic folding ability of the talin domain as compared to the absence of any chaperones and thus, the intersection force overlaps closely with that of the control. Interestingly, DsbA as a foldase chaperone shifts the free energy landscape towards the folded state by increasing the free energy barrier, which in turn increases the stability of the folded state. This signifies DsbA assists the talin folding at higher forces, increasing the half-point force and thus, could remain folded still at 14.4 pN. Interestingly, we found the distance to the transition state remains unchanged, signifying that the chaperones does not affect the mechanical rigidity of the native substrate. This chaperone-modulated folding dynamics eventually affects the talin mechanical stability which has been reconciled from the loading rate experiment. We found that the unfolding force at zero force has been observed to decrease with DnaK (unfoldase) and increase in the presence of DsbA (foldase). During the unfoldase interaction with talin, it is likely that the unfoldase chaperones limit the conformational space of the extended substrate to prevent them from misfolding and aggregation. Similarly, hydrophobic side chains of DnaK/Hsp70 substrate binding-cavity could form van der Waals interactions with the side chains of the mechanically extended substrate, which could suggest DnaK/Hsp70 affinity for the hydrophobic peptide segments. However, in the DnaKJ complex, DnaJ and extended substrate accelerate the ATP hydrolysis by DnaK chaperone, which triggers the conformational changes of the DnaK lid. This leads to the non-covalent substrate entrapment by the DnaK chaperone.

2) This might not be completely critical, but it would be useful to perform at least limited test measurements on whether the observed trends also hold up with the wild-type R3 domain, in addition to the IVVI mutant used.

We appreciate the comment of reviewer#2. To understand the effect of chaperones on wild type (WT) R3 domain, we measured the intersection force of R3-WT in the presence of different chaperones and observed similar pattern that we found with mutant protein. Since WT-R3 is mechanically weaker than the R3-IVVI form, they exhibit intersection force at 6.3 ± 0.3 pN. We observed holdase and foldase chaperones can modulate this force by altering the folding dynamics. For example, in the presence of a holdase chaperone such as DnaK, the intersection force is decreased to 4.4 ± 0.2 pN, while upon the addition of foldase chaperone DsbA, it increases to 9.8 ± 0.3 pN. For the DnaKJE complex, the intersection force of R3-WT has been measured to be 6.3 ± 0.2 pN (Figure 4A). Furthermore, we tested the statistical significance of different chaperones data sets by one way ANOVA analysis, followed by their mean comparison by Bonferroni test at $*p \leq 0.05$ level. We observed that the differences are statistically significant for different chaperone pair sets. However, DnaJ and DnaK pair, both

acting as unfoldases, exhibit no such significant mean comparison. Similarly, control-DnaKJE pair exhibits non-significant mean difference as DnaKJE only promotes innate folding ability in talin, as compared to the control and shows no such observable effect (Figure 4B and 4C). For the convenience of reviewer#2, we have provided the figure below and also included the figure in the revised manuscript.

Figure 4: Intersection force of talin WT-R3: (A) Intersection force: Intersection forces of talin WT-R3 are measured both in the absence of any chaperones (control) and in the presence of the different chaperones. Intersection force is lowered in case of unfoldase chaperones such as, DnaJ and DnaK, while increases with the foldase such as, DsbA. Data points are calculated using four individual molecules. Error bars are represented as s.e.m. **(B) ANOVA analysis:** ANOVA analysis with different chaperones (n=4 molecules for each population) shows that intersection force for each chaperone population are statistically significant (R-sq. = 0.96) at $*p \leq 0.05$ level, with lower force in the presence of unfoldase chaperones and higher forces in the presence of DsbA (foldase). Since the DnaKJE complex promotes native folding ability in the substrate as in control, talin intersection forces are the same both in the control and with the DnaKJE complex, showing non-significant differences (ns). Similarly, unfoldases- DnaJ and DnaK also exhibit non-significant differences in the intersection forces. **(C) Bonferroni post-hoc test:** Furthermore, we performed Bonferroni post-hoc test to check the pair comparison and observed that the means differences are statistically significant except control-DnaKJE pair and DnaJ-DnaK pairs.

3) The key results of the paper are a number of unfolding/refolding force values. These appear to be reported as means +/- standard error of the mean? Is this what was done? It should be explicitly stated (e.g. on page, when the first values are reported in the form $x \pm y$) how the values were generated. Importantly, rupture forces at a given loading rate follow a Bell-Evans distribution, not a Gaussian distribution. This should be considered when analyzing the data.

We appreciate the reviewer for the comment. Reviewer#2 is absolutely correct that the unfolding/refolding force values have been reported as mean \pm standard error of the mean and we have explicitly stated these values at the first place in the manuscript. We also agree with the reviewer that rupture force follows a Bell-Evans distribution and have mentioned it in the revised manuscript.

4) The author state "Protein exhibits force-independent unfolding transition due to the rigid folded state, and thus, the unfolding kinetics fits better to Bell-like equation." This is unclear/incorrect for several reasons

--- Folding is certainly NOT force-independent! The authors show themselves how for their system the protein unfolds under force, with a rate that increases with force (Figure 4).

--- The Bell equation is precisely used to quantify the force dependence!

--- The statement “fits better” is unclear – better than what?

We apologize to the reviewer for the mistake in the statement. We have corrected it to “force-dependent unfolding transition” in the revised manuscript. Reviewer#2 is absolutely correct that Bell equation is certainly used to quantify the force-dependency of the folding/unfolding rates.

We also appreciate the comment of the reviewer regarding the Bell equation. We wanted to mention that Bell-equation could fit better to the unfolding kinetics than the refolding kinetics, as the collapse-associated energetics of unfolded polypeptide has a large contribution to determining the observed folding kinetics (Guo et al., Chem Sci., 2018 and Berkovich et al., Biochem. Biophys. Res. Commun., 2010). Unlike unfolding transition state distance, refolding transition state distance is highly force-dependent due to compliance variation of the unfolded state (defined as the origin state of the folding transition) as well as the effect of tether to extract the precise kinetic parameters. Unfolded states are highly flexible polypeptide chains and thus, are easily deformed by pulling device. Because of this deformation, the folding transition faces an extra contribution to the refolding barrier height, which is regulated by the ratio of $K_s/K_u(F)$, where K_s is the stiffness of the pulling device and $K_u(F)$ is the stiffness of unfolded state (Please see Pierse et al., Biophys J., 2013 for in detail discussion). Thus, the approximation of the applied force may not be precise for the flexible unfolded state or even soft intermediate like transition state (Zhang & Dudko et al., PNAS 2013). Notably, talin as an α helical protein is mechanically compliant and its unfolded basin could be changed non linearly with the force (Petrosyan et al., JMB., 2021; Hoffmann et al., Chem. Soc. Rev., 2012), which is also evident in the two-state substrate (Stannard et al., Nano Lett., 2021). The nature of this dynamic refolding transition is strongly attributed to the varying entropic elasticity of the extended polyprotein (or denatured polymer) under different forces within the 6-10 pN range (Yao et al., Nat. Commun., 2016; Berkovich et al., Biochem. Biophys. Res. Commun., 2010; Eckels et al., Annu. Rev. Physiol., 2018; Eckles et al., Cell Rep., 2019; Tapia-Rojo et al., BioRxiv, 2021). This results in non-linear force-dependency on the logarithmic scale, and therefore, Bell model is not a good approximation for the folding kinetics (Chen et al., JACS 2015, Yao et al., 2016). Here we used the Bell-like equation to extract kinetic parameters due to its simplicity (please see page 2880 on Li et al., Chem Sci., 2021). Thus, the refolding transition state distance becomes asymmetric due to the substantial change in the unfolding state position, which becomes more complicated with the presence of different chaperones.

5) I think it is good that the authors, in addition to studying prokaryotic chaperones that would not likely interact with talin, also eukaryotic chaperones Hsp70 and Hsp40. Still their statement “Since these chaperones reside in eukaryotic cytosol, they most likely to interact with talin” is an overstatement. Many things are in the cytosol and this does not mean they all interact!

Reviewer#2 is absolutely correct. As we are not sure whether the cytoplasmic chaperones interact with talin or not, we have modified the statement in the revised manuscript.

6) How were the free energy differences computed? From the fits of the rates by Equations 2 and 3? Doesn't the “molecular attempt frequency” A influence the results? How well is this known? Do the results match the free energy differences suggested by the equilibrium measurements via $\text{Energy} = [\text{midpoint force}] \times [\text{distance between folded and unfolded state}]$?

We thank the reviewer for the comment. Reviewer#2 is absolutely correct that we determined the free energy values from Equations 2 and 3. We plotted the \ln of unfolding and refolding rates at different forces and fitted to the Bell-like equation as previously reported by different groups (Kuo et al., PNAS 2010; Yao et al., Nat Commun. 2016).

We also agree with the reviewer that molecular attempt frequency (A) influences the results through changing the force-free rate constant, k_0 , which is calculated as $k_0 = A * e^{-\Delta G_0/k_B T}$ and therefore, allow us to calculate the absolute value of the free energy barrier at zero force condition. Fitting \ln unfolding values with Bell-like equation using the A value of 10^6 s^{-1} , gives an average barrier height within the range of 20-30 $k_B T$. This A value has been previously reported by different groups (Kuo et al., PNAS 2010; Schuler et al., Nature 2002; Garcia-Manyes et al., PNAS 2009). As the reviewer pointed out, A value could be changed within a wide range of 10^6 to 10^{13} s^{-1} (Carrion-Vazquez et al., PNAS. 1999; Dougan et al., PNAS. 2008; Garcia-Manyes et al., PNAS. 2009). However, a value of 10^6 s^{-1} is assumed to be practically relevant for attempt frequency based on the polymer scaling law and diffusion theory (Li et al., 2004; Kubelka et al., 2004). Attempt frequency is primarily changed due to the type of the reactive system such as proteins and reaction occurrence in solution, where it has been observed to decrease significantly than predicted by transition state theory (Garcia-Viloca et al., Science 2004). In force spectroscopy approaches, the kinetics data are estimated either by Monte-Carlo simulations or by fitting to different mathematical framework such as Bell-like, Cusp and linear-cubic model which affects the attempt frequency value (Garcia-Manyes et al., PNAS. 2009; Carrion-Vazquez et al., PNAS. 1999; Schlierf et al., PNAS. 2004). It has been estimated that the A value is higher (10^6 s^{-1}) in the Bell-like equation than both the cusp and linear-cubic model (10^2 s^{-1}) due to transition state dependency while calculating the reaction rates (Garcia-Manyes et al., PNAS. 2009). Our obtained range of barrier height is in well-agreement to previously reported value of barrier height by Löff et al., (PNAS 2019). However, the A value is likely to differ from the values estimated by other force spectroscopic studies. This is due to the different reaction co-ordinates in force spectroscopic approaches, where the change in protein extension is probed with the experimental timescale and different folding mechanism underlying the mechanical unfolding process.

As suggested by the reviewer, we have also calculated the energy barrier height by equilibrium energy measurement and observed that the barrier values are 45.3 $k_B T$ in control, which decreases to 33.3 $k_B T$ with Hsp70 and increases to 66.1 $k_B T$ with DsbA. Notably, the changing pattern of unfolding barrier values is the same as obtained by Bell-like equation. We have included this section in the revised manuscript.

7) I am not sure I understand the point the authors are trying to make in the Discussion (page 9), when they compare their energies to the literature. The authors find folding free energies in the range of 20-30 kT, and changes by the chaperones of a few kT. However, they then go on to cite a number of references where the folding free energy was on the order of a few kT. I do not doubt that the results presenting in the paper are in a plausible range of values, but wouldn't more relevant references be cases where the actual folding free energy is on the order of 20-30 kT and then external factors change that number by a few kT? One example also using magnetic tweezers is e.g.

Multiplexed protein force spectroscopy reveals equilibrium protein folding dynamics and the low-force response of von Willebrand factor.

Löff A, Walker PU, Sedlak SM, Gruber S, Obser T, Brehm MA, Benoit M, Lipfert J. Proc Natl Acad Sci U S A. 2019 Sep 17;116(38):18798-18807. doi: 10.1073/pnas.1901794116.

where [midpoint force] x [distance between folded and unfolded state] appear to be in the range of 20-30 kT, which changes of a few kT due to conditions.

Reviewer#2 is absolutely correct that the references, demonstrating the folding free energy of 20-30 kT are more relevant in understanding the chaperone-modulated folding free energy. We have cited the references in the revised manuscript.

8) Given the many pairwise comparisons of forces in the manuscript, it would be good to compute statistical significances for the differences and to also report them in the figures (e.g. in Figure 2)

We appreciate reviewer#2 comment on the statistical significance of the data and we have included it in the revised manuscript.

We have performed the ANOVA (one-way) analysis of both unfolding and refolding force of talin with different chaperones and checked their pair comparison by Bonferroni post-hoc test at $*p \leq 0.05$ level (Fig. 5 and 6). We observed that the unfolding force changes with concentrations for different chaperones; however, the changes are not significant in all the cases. Since this mechanical effect becomes saturated at a particular chaperone concentration, the unfolding force remains unchanged and thus, no significant differences are observed beyond that concentration. For example, with DnaJ, the unfolding force becomes saturated at 1 μM concentrations and no significant differences are observed upon increasing the concentration to 3 μM (Fig. 5A and 5B). Similarly, for DnaK, no such significant difference is observed upon increasing the concentration from 3 to 10 μM concentrations (Fig. 5C and 5D). Similar to unfolding force, the refolding force has also been observed to decrease and become saturated at 1 μM DnaJ and 3 μM DnaK and exhibit same statistical significance (Fig. 6A to 6D). Though it has been observed that both the unfolding and refolding force decrease significantly with the DnaKJ complex, the complex is expected to possess the holdase activity, which is relieved upon GrpE addition (Fig. 5E, 5F and 6E, 6F). Additionally, for Hsp40 and Hsp70, we have performed one way ANOVA analysis to check the statistical significance of the force data sets and observed that both the unfolding and refolding forces are significantly different lower than that of control data sets (Fig. 7).

Figure 5: One-way ANOVA analysis of talin unfolding force with different chaperones: (A and B) DnaJ: ANOVA analysis has been performed to check the statistical significance of unfolding force with different concentrations of DnaJ. We observed that the unfolding force becomes saturated at 1 μM , and thus the mean force differences become non-significant (ns) between 1 and 3 μM concentrations at $*p \leq 0.05$ level ($R \text{ sq.} = 0.456$). We further performed the Bonferroni post hoc test to check the mean comparisons with different DnaJ concentrations. Similarly, we performed the ANOVA analysis for **(C and D) different DnaK concentrations** and found that the unfolding force becomes saturated at 3 μM and thus showed non-significance between 3 and 10 μM concentration ($R \text{ sq.} = 0.454$ at $*p \leq 0.05$). **(E and F) DnaKJ and DnaKJE:** Unfolding force with DnaKJ is statistically significant with both DnaKJE and control ($R \text{ sq.} = 0.184$), which is also evident in post-hoc analysis. **(G and H) DsbA:** The unfolding force in the absence of DsbA (control) is significantly different from 60 μM DsbA ($R \text{ sq.} = 0.670$), which is shown in the Bonferroni test. For each chaperone data set, unfolding forces are measured and averaged. For Control, $n=15$; DnaJ, $n=10$ (0.5 μM); $n=16$ (1 μM), $n=17$ (3 μM); DnaK, $n=8$ (1 μM); $n=17$ (3 μM), $n=28$ (10 μM); DnaKJ, $n=15$; DnaKJE, $n=12$; DsbA, $n=13$ (30 μM), $n=8$ (60 μM).

Figure 6: One-way ANOVA analysis of talin refolding force with different chaperones: (A and B) DnaJ: ANOVA analysis has been performed to check the statistical significance of refolding force at different concentration of DnaJ. We observed that the refolding force becomes saturated at 1 μM , and thus the mean force differences become non-significant between 1 and 3 μM concentrations at $*p \leq 0.05$ level ($R \text{ sq.} = 0.363$). We further performed the Bonferroni post-hoc test to check the mean comparison with different DnaJ concentrations. Similarly, we performed the ANOVA analysis for **(C and D)** different DnaK concentrations and found that the refolding force becomes saturated at 3 μM and thus showed non-significance between 3 and 10 μM concentration ($R \text{ sq.} = 0.414$ at $*p \leq 0.05$). **(E and F) DnaKJ and DnaKJE:** Refolding force with DnaKJ is statistically significant with both DnaKJ and control ($R \text{ sq.} = 0.297$), which is also evident in post-hoc analysis. **(G and H) DsbA:** The refolding force in the absence of DsbA (control) is significantly lower from 60 μM DsbA ($R \text{ sq.} = 0.703$), which is shown in Bonferroni post-hoc test. For each chaperone data set, refolding force are measured and averaged. For Control, $n=16$; DnaJ, $n=11$ (0.5 μM); $n=16$ (1 μM), $n=24$ (3 μM); DnaK, $n=7$ (1 μM); $n=14$ (3 μM), $n=23$ (10 μM); DnaKJ, $n=11$; DnaKJE, $n=14$; DsbA, $n=10$ (30 μM), $n=10$ (60 μM).

Figure 7: Mechanical stability of talin with Hsp40 and Hsp70: Hsp70 and Hsp40 chaperones decrease the mechanical stability of talin by changing their unfolding and refolding force. **(A) Unfolding force:** In the absence of any chaperones (control), the unfolding force of talin is 10.8 ± 0.2 pN, which has been observed to decrease to 8.3 ± 0.4 pN and 7.7 ± 0.5 pN with Hsp40 and Hsp70, respectively. Data points are measured using more than five individual molecules with more than 12 unfolding events. Error bars represent standard error of mean (s.e.m.). **(D) Refolding force:** Similarly, the refolding force has been observed to shift from 10 ± 0.3 pN (control) to 7 ± 0.5 and 6.8 ± 0.6 pN with Hsp40 and Hsp70, respectively. Data points are measured by averaging minimum three molecules. Error bars are represented as s.e.m. **ANOVA analysis of (B and C) unfolding force and (E and F) refolding force:** To check the statistical significance of the force data sets with Hsp40 and Hsp70 chaperones, we have also

performed one-way ANOVA analysis, followed by their pair comparison by Bonferroni post-hoc test at $*p \leq 0.05$ level. It has been observed that both the unfolding and refolding forces are significantly lower than that of control. Since Hsp40 and Hsp70 act as unfoldases, their mean force differences are non-significant. $R\text{-sq.} = 0.630$ for unfolding and 0.662 for refolding. For unfolding force analysis, $n = 15$ (Control); $n = 12$ (Hsp40); $n = 7$ (Hsp70). For refolding force, $n = 16$ (Control); $n = 12$ (Hsp40); $n = 6$ (Hsp70).

Minor points:

- I would probably be good to write out/define "RIAM" in the Abstract.

We thank the reviewer for the suggestion. We have changed this in the revised manuscript.

- In the Abstract, it would help to state also the midpoint force in the absence of chaperones. It is stated that "holdase chaperones reduce their unfolding force to ~6 pN", but one wonders what the force was before this intervention.

We thank the reviewer for the suggestion. We have included the force value in the revised manuscript.

- The authors emphasize the "real-time" nature of their measurements, e.g. in the title of the paper and twice in one sentence on page 4. Is this really so relevant? Having time resolution is important, but it seems to me that recording the time traces and post-processing them would provide exactly the same information as real-time processing?

We understand the point of the reviewer. We have removed the "real-time" term from the title of the revised manuscript.

- "Equation" or "Eq." should be capitalize when referring to the specific equation.

We thank the reviewer for pointing this out and we have changed this in the revised manuscript.

- Something is missing in "Magnetic tweezers technology, using fore-clamp protocol, allows us".

We thank the reviewer for pointing this out and we have changed this in the revised manuscript.

We thank again reviewer#2 for the valuable comments and suggestions, which have immensely improved the quality of our manuscript.

REVIEWERS' COMMENTS:

Reviewer #1 (Remarks to the Author):

The authors Chakraborty et al. complemented the manuscript "Real time observation of chaperone-modulated talin mechanics under single molecule resolution" with new data and discussions. The inhibition experiments for Hsp70 and Hsp40, as well as the modifications in response to the other reviewer's comments, significantly improve the work.

I understand the argumentation the authors make regarding the experiments in cells (e.g. fibroblasts) and the potential side effects of Hsp70 inhibitors on other focal adhesion proteins. In fact, my suggestion was meant positively. It would be very nice if, in the future, the authors find a way to show a specific effect of the chaperones on talin mechanics in a cellular model. For the moment being, keeping the discussion as a "plausible in vivo effect" is fine.

Minor remarks:

- Fig. 1) Use the same colour code for bars and boxplots, otherwise it can be confusing, e.g. for the bars (A-C) the Hsp70 is blue, for the boxplots instead the Hsp+I. The same should be done also for Fig. 2 and 4.
- The SDS-PAGE images of protein samples (Fig. 3) are fine, no additional Western blots needed.

Reviewer #2 (Remarks to the Author):

Chakraborty et al. have addressed all my comments during the revision of their manuscript and I feel that the work is ready for publication.

REVIEWERS' COMMENTS:

Reviewer #1 (Remarks to the Author):

The authors Chakraborty et al. complemented the manuscript “Real time observation of chaperone-modulated talin mechanics under single molecule resolution” with new data and discussions. The inhibition experiments for Hsp70 and Hsp40, as well as the modifications in response to the other reviewer's comments, significantly improve the work.

I understand the argumentation the authors make regarding the experiments in cells (e.g. fibroblasts) and the potential side effects of Hsp70 inhibitors on other focal adhesion proteins. In fact, my suggestion was meant positively. It would be very nice if, in the future, the authors find a way to show a specific effect of the chaperones on talin mechanics in a cellular model. For the moment being, keeping the discussion as a “plausible in vivo effect” is fine.

We would like to thank reviewer#1 for the suggestions, which have substantially improved the quality of the manuscript.

Minor remarks:

- Fig. 1) Use the same colour code for bars and boxplots, otherwise it can be confusing, e.g. for the bars (A-C) the Hsp70 is blue, for the boxplots instead the Hsp+I. The same should be done also for Fig. 2 and 4.

We thank the reviewer for the comment and we have changed the colour code of the figures in the revised manuscript.

- The SDS-PAGE images of protein samples (Fig. 3) are fine, no additional Western blots needed.

We thank the reviewer for the comment.

Reviewer #2 (Remarks to the Author):

Chakraborty et al. have addressed all my comments during the revision of their manuscript and I feel that the work is ready for publication.

We would like to thank reviewer#2 for the comments.